# On the Interpolation Effect of Score Smoothing in Diffusion Models

**Zhengdao Chen**
Google Research
Mountain View, CA 94043, USA
`zhengdao.c3@gmail.com`

## Abstract

Diffusion models have achieved remarkable progress in various domains with an intriguing ability to produce new data that do not exist in the training set. In this work, we study the hypothesis that such creativity arises from the neural network backbone learning a smoothed version of the empirical score function, which guides the denoising dynamics to generate data points that interpolate the training data. Focusing mainly on settings where the training set lies uniformly in a one-dimensional subspace, we elucidate the interplay between score smoothing and the denoising dynamics with analytical solutions and numerical experiments, demonstrating how smoothing the score function can cause the denoised data samples to interpolate the training set along the subspace. Moreover, we present theoretical and empirical evidence that learning score functions with neural networks - either with or without explicit regularization - can naturally achieve a similar effect, including when the data belong to simple nonlinear manifolds. [1]

## 1 Introduction

Score-based diffusion models (DMs) have become an important pillar of generative modeling across a variety of domains from content generation to scientific computing (Sohl-Dickstein et al., 2015; Song and Ermon, 2019; Ho et al., 2020; Ramesh et al., 2022; Abramson et al., 2024; Brooks et al., 2024). After being trained on datasets of actual images or molecular configurations, for instance, such models can transform noise samples into high-quality images or chemically-plausible molecules that do not belong to the training set, indicating an exciting capability of such models to generalize beyond what they have seen and, in a sense, be creative.

How such creativity can arise in score-based DMs is an intriguing theoretical question. At the core of these models is the training of neural networks (NNs) to fit a series of target functions, often called the *empirical score function (ESF)*, which drive the *denoising process* at inference time. The precise form of the ESF is determined by the training set and can in principle be computed exactly, but when equipped with the exact ESF instead of the NN-learned version, the DM will end up generating data points that already exist in the training set (Yi et al., 2023; Li et al., 2024a; Kamb and Ganguli, 2025), a phenomenon commonly called *memorization*. This suggests that, for DMs to generate fresh samples beyond the training set, it is crucial that the NN does *not* learn the ESF perfectly. But what exactly is the *desirable* kind of "imperfection" here, and how can it give rise to the creativity of DMs?

Our work takes inspiration from an interesting proposal by Scarvelis et al. (2025) that smoothing the score function can make a DM generate samples at barycenters of the training data. In this work, we argue that regularization effects in NN training *naturally* cause it to learn a smoothed version of the ESF. Then, we show quantitatively how a smoothed ESF leads the denoising dynamics to generate samples that interpolate the training data along the data subspace, which can be considered a simplest form of meaningful creativity. Our main contributions are as follows:

1. We show theoretically that regularized two-layer ReLU NNs tend to approximately learn a smoothed version of the ESF (named the "Smoothed PL-ESF"). We focus on a simple setup

---

[1]Code available at https://github.com/google-research/diffusion-score-smoothing.

with uniformly-spaced training data and prove the result by analyzing a non-parametric variational problem involving the score matching loss and a non-smoothness penalty known as equivalent to NN regularization;

2. Through analytical solutions of the denoising dynamics, we show that a DM equipped with the Smoothed PL-ESF produces a non-singular density that interpolates the training set. In particular, when the training data lie in a one-dimensional (1-D) subspace, the denoising dynamics is able to recover the underlying subspace without collapsing onto the training set.

3. Through numerical experiments, including in multi-dimensional and nonlinear settings, we validate empirically that (1) neural networks tend to learn smoothed versions of the ESF either with or without explicit regularization in training, and (2) this score smoothing effect leads the denoising dynamics to generate samples that interpolate the training data.

Together, these results shed light on how score smoothing can be an important causal link for understanding how NN-based DMs avoid memorization.

The rest of the paper is organized as follows. After briefly reviewing the background in Section 2, we examine the smoothing of ESF in the 1-D case and discuss its connections with NN regularization in Section 3. The trajectory of the denoising dynamics under the Smoothed PL-ESF is derived in Section 4. In Section 5, we generalize the analysis to the multi-dimensional case when the training data belongs to a hidden subspace. In Section 6, we present empirical evidence that NN-learned SF exhibits an interpolation effect similar to that of score smoothing, including when the data belongs to a nonlinear manifold. We defer the discussion of related works to Appendix A.

**Notations** For $x, \delta > 0$, we write $p_{\mathcal{N}}(x; \sigma) = (\sqrt{2\pi}\sigma)^{-1} \exp(-x^2/(2\sigma^2))$ for the 1-D Gaussian density with mean zero and variance $\sigma^2$, $\boldsymbol{\delta}_x$ for the Dirac delta distribution centered at $x \in \mathbb{R}$, and $\mathrm{sgn}(x)$ for the sign of $x$. We write $[n] := \{1, .., n\}$ for $n \in \mathbb{N}_+$. For a vector $\boldsymbol{x} = [x_1, ..., x_d] \in \mathbb{R}^d$, we write $[\boldsymbol{x}]_i = x_i$ for $i \in [d]$. The use of big-O notations is explained in Appendix B.

## 2 BACKGROUND

Score-based DMs have many variants, and we will focus on a simple one (named the "Variance Exploding" version by Song et al. 2021c) where the *forward (or noising) process* is defined by the following stochastic differential equation (SDE) in $\mathbb{R}^d$ for $t \geq 0$:

$$d\mathbf{x}_t = d\mathbf{w}_t , \quad \mathbf{x}_0 \sim p_0 , \tag{1}$$

where $\mathbf{w}$ is the Wiener process (a.k.a. Brownian motion) in $\mathbb{R}^d$. The marginal distribution of $\mathbf{x}_t$, denoted by $p_t$, is thus fully characterized by the initial distribution $p_0$ together with the conditional distribution, $p_{t|0}(\boldsymbol{x}|\boldsymbol{x}') = \prod_{i=1}^{d} p_{\mathcal{N}}([\boldsymbol{x}]_i - [\boldsymbol{x}']_i; \sqrt{t})$: specifically, $p_t$ is obtained by convolving $p_0$ with an isotropic Gaussian distribution with variance $\sigma(t)^2 = t$ in every direction.

A key observation is that this process is equivalent (in marginal distribution) to a deterministic dynamics, often called the *probability flow ordinary differential equation (ODE)* (Song et al., 2021c):

$$d\mathbf{x}_t = -\tfrac{1}{2}\boldsymbol{s}_t(\mathbf{x}_t)dt , \tag{2}$$

where $\boldsymbol{s}_t(\mathbf{x}) = \nabla \log p_t(\mathbf{x})$ is the *score function (SF)* associated with the distribution $p_t$ (Hyvärinen and Dayan, 2005) and also fundamentally connected to the denoiser function via Tweedie's formula (Efron, 2011). In generative modeling, $p_0$ is often a distribution of interest that is hard to sample directly (e.g. the distribution of cat images in pixel space), but when $T$ is large, $p_T$ is close to a Gaussian distribution (with variance increasing in $T$), from which samples are easy to obtain. Thus, to obtain samples from $p_0$, we may first sample from $p_T$ and follow the *reverse (or denoising) process* by numerically solving (2) backward-in-time (or its equivalent stochastic variants, which we will not focus on; see Song et al., 2021a;c). A main challenge in this procedure lies in the estimation of the family of SFs, $\nabla \log p_t$ for $t \in [0, T]$. In reality, we have no prior knowledge of each $p_t$ (or even $p_0$) but just a training set $S = \{\boldsymbol{y}_k\}_{k \in [n]}$ usually assumed to be sampled from $p_0$. Thus, we only have access to an *empirical* version of the noising process, where the same SDE (1) is initialized at $t = 0$ with not $p_0$ but the uniform distribution over $S$ (i.e., $\mathbf{x}_0 \sim p_0^{(n)} := \frac{1}{n} \sum_{k=1}^{n} \delta_{\boldsymbol{y}_k}$), and hence the marginal distribution of $\mathbf{x}_t$ is $p_t^{(n)}(\boldsymbol{x}) := \frac{1}{n} \sum_{k=1}^{n} p_{t|0}(\boldsymbol{x}|\boldsymbol{y}_k)$, called the *noised empirical*

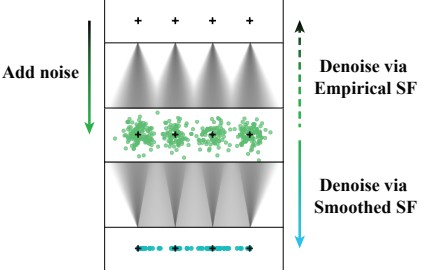

Figure 1: From the noised empirical distribution ($p_{t_0}^{(n)}$; **middle**), denoising with the ESF ($\nabla \log p_t^{(n)}$) leads back to the empirical distribution of the training set ($p_0^{(n)}$; **top**), while using a smoothed SF (e.g. the Smoothed PL-ESF, $\hat{s}_{t,\delta_t}^{(n)}$; or NN-learned SF) produces a distribution that interpolates among the training set on the relevant subspace (e.g., $\hat{p}_0^{(n,t_0)}$ in the case of Smoothed PL-ESF; **bottom**). Definitions are given in Sections 2 - 4.

*distribution* at time $t$. To obtain a proxy for $\nabla \log p_t$, one often uses an NN as a (time-dependent) score estimator, $s_{\boldsymbol{\theta}}(\boldsymbol{x}, t)$, and train its parameters to minimize variants of the time-averaged *score matching loss* (Song et al., 2021c):

$$\min_{\boldsymbol{\theta}} \frac{1}{T} \int_0^T L_t^{(n)}[\boldsymbol{s}_{\boldsymbol{\theta}}(\,\cdot\,, t)]dt \;, \tag{3}$$

where

$$L_t^{(n)}[\boldsymbol{f}] := t \cdot \mathbb{E}_{\mathbf{x} \sim p_t^{(n)}} \left[ \left\| \boldsymbol{f}(\mathbf{x}) - \nabla \log p_t^{(n)}(\mathbf{x}) \right\|^2 \right] \tag{4}$$

measures the $L^2$ distance between the score estimator and $\nabla \log p_t^{(n)}$ — which is the *ESF* at time $t$ — with respect to $p_t^{(n)}$. The scaling factor of $t \propto 1/\mathbb{E}[\nabla \log p_{t|0}(\mathbf{x}_t|\mathbf{x}_0)]$ serves to balance the contribution to the loss at different $t$ (Song et al., 2021c).

Though in practice the minimization problem (3) is solved via NN optimization and Monte-Carlo sampling (Vincent, 2011), we know the minimum is attained uniquely by the ESF itself, which can be computed in closed form (e.g. see Section 3). If we use the ESF directly in the denoising dynamics (2) instead of an NN-learned SF, we get an empirical version of the probability flow ODE:

$$d\mathbf{x}_t = -\tfrac{1}{2} \nabla \log p_t^{(n)}(\mathbf{x}_t) \;, \tag{5}$$

which exactly reverses the empirical forward process of adding noise to the training set, and hence the outcome at $t = 0$ is inevitably $p_0^{(n)}$. In other words, the model *memorizes* the training data. This suggests that the creativity of the diffusion model hinges on a *sub-optimal* solution to the minimization problem (3) and an *imperfect* approximation to the ESF. Indeed, the memorization phenomenon has been observed in practice when the models have large capacities relative to the training set size (Gu et al., 2025; Kadkhodaie et al., 2024), which results in too good an approximation to the ESF. This leads to the hypothesis that regularized score estimators — in particular, those that learn to smooth the ESF — give rise to the model's ability to generalize beyond the training set.

In Sections 3 - 4, we provide theoretical evidence for this hypothesis in the 1-D setting. We first argue that NN regularization biases the score function towards a smoothed version of the ESF by analyzing a variational problem involving the score matching loss and the non-smoothness measure associated with the weight norm of two-layer ReLU NNs. Then, we show analytically how the smoothed score drives the denoising dynamics to generate distributions that interpolate between instead of memorizing the training data.

## 3 SCORE SMOOTHING IN ONE DIMENSION

Let us begin with the simplest setup where $d = 1$ and $S = \{y_1 = -1, y_2 = 1\}$ consists of $n = 2$ points (whereas in Appendix C, we prove extensions of all results in Sections 3 - 5 to the setup where $S$ consists of $n$ uniformly-spaced points). At time $t$, the noised empirical distribution is $p_t^{(n)}(x) = \frac{1}{2}(p_{\mathcal{N}}(x + 1; \sqrt{t}) + p_{\mathcal{N}}(x - 1; \sqrt{t}))$, and the (scalar-valued) ESF takes the form of

$$\tfrac{d}{dx} \log p_t^{(n)}(x) = (\hat{x}_t^{(n)}(x) - x)/t \;, \tag{6}$$

where

$$\hat{x}_t^{(n)}(x) := \mathbb{E}_{0|t}[\mathbf{x}_0|\mathbf{x}_t = x] = \frac{p_{\mathcal{N}}(x-1; \sqrt{t}) - p_{\mathcal{N}}(x+1; \sqrt{t})}{p_{\mathcal{N}}(x-1; \sqrt{t}) + p_{\mathcal{N}}(x+1; \sqrt{t})} \tag{7}$$

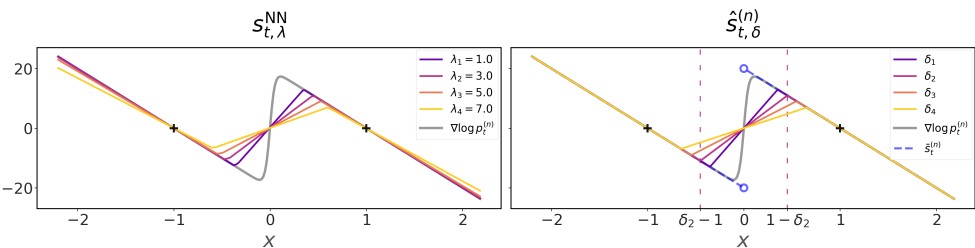

Figure 2: Similarities between NN-learned SF ($s_{t,\lambda}^{\text{NN}}$) under increasing strengths of regularization, $\lambda$ (**left**) and the Smoothed PL-ESF ($\hat{s}_{t,\delta}^{(n)}$) with decreasing values of $\delta$ (**right**) in the $d = 1$, $n = 2$ case with a fixed $t$. Details of the experiment setup are discussed in Section J.1.

lies between $\pm 1$ and has the same sign as $x$. As $t \to 0$, the Gaussians sharpen and $\hat{x}_t^{(n)}(x)$ approaches $\text{sgn}(x)$, allowing us to approximate the ESF by a piece-wise linear (PL) function, named the *PL-ESF*:

$$\bar{s}_t^{(n)}(x) = (\text{sgn}(x) - x)/t \,, \tag{8}$$

which explains the attraction of the backward dynamics to $\pm 1$, corresponding to memorization. We will show below that this behavior can crucially be avoided by smoothing the ESF at small $t$.

### 3.1 SCORE SMOOTHING VIA NN REGULARIZATION: A THEORETICAL MODEL

To gain intuition, we perform experiments to fit the ESF (6) at a fixed $t$ using the loss (4) by two-layer ReLU NNs that are regularized by weight decay (additional details given in Appendix J.1). As shown in Figure 2, for various strengths of regularization (denoted by $\lambda$), the NN-learned score estimators are nearly PL, and remarkably, well-approximated by the following ansatz parameterized by $\delta \in (0, 1]$:

$$\hat{s}_{t,\delta}^{(n)}(x) := \begin{cases} -(x+1)/t \,, & \text{if } x \leq \delta - 1 \,, \\ -(x-1)/t \,, & \text{if } x \geq 1 - \delta \,, \\ \delta/(1-\delta) \cdot x/t \,, & \text{if } x \in (\delta - 1, 1 - \delta) \,. \end{cases} \tag{9}$$

In particular, a stronger regularization corresponds to a smaller $\delta$. We will refer to $\hat{s}_{t,\delta}^{(n)}$ as a *Smoothed PL (S-PL) ESF*. As illustrated in Figure 2 (right), it is PL and matches $\bar{s}_t^{(n)}$ except on the interval $[\delta - 1, 1 - \delta]$ (hence $\hat{s}_{t,1}^{(n)} \equiv \bar{s}_t^{(n)}$). In Appendix I.1, we show that (9) can be obtained by averaging the PL-ESF over local windows.

Why do the regularized NNs learn score estimators that are so close to being expressed by (9)? While it is difficult to predict exactly what function an NN will learn due to the nonlinearity and stochasticity of the training dynamics, below we will provide an argument based on a non-parametric view of NN regularization. Specifically, for function fitting in one dimension, it is shown in Savarese et al. (2019) that regularizing the weight norm of a two-layer ReLU NN (with unregularized bias and linear terms) – equivalent to applying weight decay during training – is essentially equivalent to penalizing a *non-smoothness* measure of the estimated function defined as:

$$R[f] := \int_{-\infty}^{\infty} |f''(x)| dx \,, \tag{10}$$

where $f''$ is the weak second derivative of the function $f$. Note that even without explicit weight norm regularization, this term has also been associated with the *implicit* regularization effect that gradient-based training induces in two-layer ReLU NNs (Chizat and Bach, 2020).

Inspired by this connection, for $\epsilon, t > 0$, we consider the following family of variational problems in function space as a proxy for NN learning:

$$r_{t,\epsilon}^* := \inf_f \quad R[f] \qquad \text{s.t.} \qquad L_t^{(n)}[f] < \epsilon \,, \tag{11}$$

with the infimum taken over all functions $f$ on $\mathbb{R}$ that are twice differentiable except on a finite set (a broad class of functions that include, e.g., any function representable by a finite-width NN).

Heuristically, we seek to minimize the non-smoothness measure among functions that are $\epsilon$-close to the ESF according to the score matching loss (3). When $\epsilon = 0$, only the ESF itself satisfies the constraint and hence attains the minimum uniquely; if $\epsilon$ is small but positive, the feasible set is infinite and we need to study how the non-smoothness penalty biases the score estimator *away from* the ESF.

Due to the non-differentiability of the functional $R$, the variational problem (11) is also hard to solve directly. Nevertheless, we can show that *near-optimality* can be achieved by the Smoothed PL-ESF uniformly across small $t$ when we choose $\delta$ to depend proportionally on $\sqrt{t}$:

**Proposition 1** *Given $\epsilon \in (0, 0.015)$, for any $\kappa \geq F^{-1}(\epsilon)$, where $F$ is a computable function that decreases strictly from 1 to 0 on $[0, \infty)$, there exists $t_1 > 0$ (dependent on $\kappa$) such that $\hat{s}_{t,\delta_t}^{(n)}$ with $\delta_t = \kappa\sqrt{t}$ satisfies the following two properties for all $t \in (0, t_1)$:*

1. *$L_t^{(n)}[\hat{s}_{t,\delta_t}^{(n)}] < \epsilon$, and hence the function $\hat{s}_{t,\delta_t}^{(n)}$ belongs to the feasible set of (11);*

2. *$R[\hat{s}_{t,\delta_t}^{(n)}] < (1 + 8\sqrt{\epsilon})r_{t,\epsilon}^*$.*

**Outline of proof** (full proof given in Appendix D): *The first property follows from the lemma below:*

**Lemma 2** *Let $\delta_t = \kappa\sqrt{t}$ for some $\kappa > 0$. Then $\exists t_1, C > 0$ (depending on $\kappa$) such that $\forall t \in (0, t_1)$,*

$$\tfrac{1}{2}F(\kappa) - C\sqrt{t} \leq L_t^{(n)}[\hat{s}_{t,\delta_t}^{(n)}] \leq \tfrac{1}{2}F(\kappa) + C\sqrt{t} \,. \tag{12}$$

*Lemma 2 (proved in Appendix E) relies on the insight that when $t$ is small, $p_t^{(n)}$ is concentrated near $\pm 1$, and hence $L_t^{(n)}$ is dominated by contributions from the neighborhood of $\pm 1$. In particular, for $L_t^{(n)}$ to remain at a constant level, we can afford to decrease $\delta_t \propto \sqrt{t}$ as $t \to 0$.*

*For the second property in Proposition 1, we observe that when $t$ is small, for any function belonging to the feasible set with a small enough $\epsilon$, its derivative near $\pm 1$ needs to be close to $d\log p_t^{(n)}/dx \approx -1/t$ (again because of the concentration of $p_t^{(n)}$ near $\pm 1$). Combined with the fundamental theorem of calculus, this gives us a lower bound on $r_{t,\epsilon}^*$.* $\qquad\square$

Proposition 1 and Lemma 2 thus establish that the Smoothed PL-ESFs with $\delta_t \propto \sqrt{t}$ are nearly minimizers of the non-smoothness measure in the function space while maintaining small bounded errors across small $t$. In Section 6, we will further show empirical evidence that when we train a single NN with regularization to learn the time-dependent SF through the time-averaged score matching loss (3), the solutions are indeed also closely approximated Smoothed PL-ESF.

## 4  INTERPOLATION EFFECT ON THE DENOISING DYNAMICS

With the motivations discussed above, we now study the effect on the denoising dynamics of substituting the ESF (6) with $\hat{s}_{t,\delta_t}^{(n)}$ where $\delta_t = \kappa\sqrt{t}$ for some $\kappa > 0$, that is, replacing (5) by:

$$\tfrac{d}{dt}\mathbf{x}_t = -\tfrac{1}{2}\hat{s}_{t,\delta_t}^{(n)}(\mathbf{x}_t) \,. \tag{13}$$

Thanks to the piece-wise linearity of (9), the backward-in-time dynamics of the ODE (13) can be solved analytically in terms of flow maps:

**Proposition 3** *For $0 \leq s \leq t < 1/\kappa^2$, the solution to (13) satisfies $\mathbf{x}_s = \phi_{s|t}(\mathbf{x}_t)$, where*

$$\phi_{s|t}(x) = \begin{cases} (1 - \delta_s)/(1 - \delta_t) \cdot x \,, & \text{if } x \in [\delta_t - 1, 1 - \delta_t] \\ \sqrt{s}/\sqrt{t} \cdot x - (1 - \sqrt{s}/\sqrt{t}) \,, & \text{if } x \leq \delta_t - 1 \\ \sqrt{s}/\sqrt{t} \cdot x + (1 - \sqrt{s}/\sqrt{t}) \,, & \text{if } x \geq 1 - \delta_t \end{cases} \tag{14}$$

The proposition is proved in Appendix G, and we illustrate the trajectories characterized by $\phi_{s|t}$ in Figure 3. The differentiability profile of $\hat{s}_{t,\delta_t}^{(n)}$ divides the $x - \sqrt{t}$ plane into three regions (**A**, **B** and **C**)

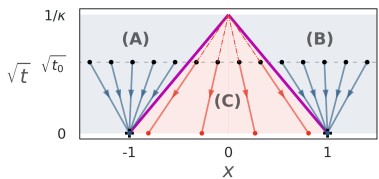

Figure 3: Phase diagram in the $x$-$\sqrt{t}$ plane for the flow solution (14) of the dynamics (13) in the $d = 1$, $n = 2$ case analyzed in Section 4.

with linear boundaries, defined by $x \leq -1 + \delta_t$, $x \geq 1 - \delta_t$ and $\delta_t - 1 \leq x \leq 1 - \delta_t$, respectively, and trajectories induced by $\phi_{s|t}$ do not cross the region boundaries. If at $t_0 > 0$, $\mathbf{x}_{t_0}$ falls into region **A** (or **B**), then as $t$ decreases to 0, it will follow a linear path in the $x - \sqrt{t}$ plane to $y_1 = -1$ (or $y_2 = 1$). Meanwhile, if $\mathbf{x}_{t_0}$ falls into region **C**, then it will follow a linear path to the $x$-axis with a terminal value between $-1$ and 1. In other words,

$$\phi_{0|t}(x) = \begin{cases} x/(1 - \delta_t), & \text{if } x \in [\delta_t - 1, 1 - \delta_t] \\ \text{sgn}(x), & \text{otherwise} \end{cases} \tag{15}$$

**Evolution of marginal distribution**   Suppose we start from some $t_0 \in (0, 1/\kappa^2)$ and run the denoising dynamics (13) backward-in-time, and we denote the marginal distribution of $\mathbf{x}_t$ by $\hat{p}_t^{(n,t_0)}$ for $t \in [0, t_0]$. We assume that $\hat{p}_{t_0}^{(n,t_0)} = p_{t_0}^{(n)}$ is the noised empirical distribution at time $t_0$.[2] Since the map $\phi_{s|t}$ is invertible and differentiable almost everywhere when $0 < s \leq t \leq t_0$, we can apply the change-of-variable formula of push-forward distributions to obtain an analytic expression for the density $\hat{p}_s^{(n,t_0)}$:

$$\hat{p}_s^{(n,t_0)}(x) = \begin{cases} (1 - \delta_t)/(1 - \delta_s) \cdot \hat{p}_t^{(n,t_0)}((1 - \delta_t)/(1 - \delta_s) \cdot x), & \text{if } x \in [\delta_s - 1, 1 - \delta_s] \\ \delta_t/\delta_s \cdot \hat{p}_t^{(n,t_0)}(\delta_t/\delta_s \cdot x + (\delta_t - \delta_s)/\delta_s), & \text{if } x \leq \delta_s - 1 \\ \delta_t/\delta_s \cdot \hat{p}_t^{(n,t_0)}(\delta_t/\delta_s \cdot x - (\delta_t - \delta_s)/\delta_s), & \text{if } x \geq 1 - \delta_s, \end{cases} \tag{16}$$

and its evolution as $s$ decreases from $t_0$ to 0 is visualized in Figure 1 (the lower grey-colored heat map). When $s = 0$, $\phi_{0|t}$ is invertible only when restricted to $[\delta_t - 1, 1 - \delta_t]$, and the terminal distribution can be decomposed as

$$\hat{p}_0^{(n,t_0)} = a_+ \boldsymbol{\delta}_1 + a_- \boldsymbol{\delta}_{-1} + (1 - a_+ - a_-)\tilde{p}_0^{(n,t_0)}, \tag{17}$$

where $a_\pm = \mathbb{E}_{\mathbf{x} \sim \hat{p}_{t_0}^{(n,t_0)}}[\mathbb{1}_{\pm\mathbf{x} \geq 1 - \delta_{t_0}}]$ and $\tilde{p}_0^{(n,t_0)}$ is a probability distribution satisfying

$$\tilde{p}_0^{(n,t_0)}(x) = \begin{cases} (1 - \delta_{t_0})/(1 - a_+ - a_-) \cdot \hat{p}_{t_0}^{(n,t_0)}((1 - \delta_{t_0})x), & \text{if } x \in [-1, 1] \\ 0, & \text{otherwise} \end{cases} \tag{18}$$

In particular, since $\hat{p}_{t_0}^{(n,t_0)}$ has a positive density on $[\delta_{t_0} - 1, 1 - \delta_{t_0}]$, $\tilde{p}_0^{(n,t_0)}$ also has a positive density on $[-1, 1]$, corresponding to a smooth interpolation between the two training data points.

Note that (18) allows us to prove KL-divergence bounds for $\hat{p}_0^{(n,t_0)}$ based on those of $\hat{p}_t^{(n,t_0)}$, and an example is given in Appendix I.2. In contrast, denoising with the exact ESF results in $p_0^{(n)}$, which is fully singular and has an infinite KL-divergence with *any* smooth density on $[-1, 1]$.

## 5   HIGHER DIMENSION: SUBSPACE RECOVERY WITH SCORE SMOOTHING

Let us consider a case where $S = \{\boldsymbol{y}_1 = [-1, 0, ..., 0], \boldsymbol{y}_2 = [1, 0, ..., 0]\} \subseteq \mathbb{R}^d$ consists of two points on the $[\boldsymbol{x}]_1$-axis (and in Appendix C we generalize the analysis to the case where $S$ contains $n$ uniformly-spaced points in any 1-D subspace). In this case, the noised empirical density is

---

[2]This can be viewed as starting from the noised empirical distribution at some large time $T$ (nearly Gaussian), initially denoising via the ESF until $t_0$, then switching to Smoothed PL-ESF for the rest of the denoising process until $t = 0$. An equivalent interpretation is that we add noise to the training data for time $t_0$ before denoising them with the Smoothed PL-ESF for the same amount (as illustrated in Figure 1).

$p_t^{(n)}(\boldsymbol{x}) = \frac{1}{2}\left(p_\mathcal{N}([\boldsymbol{x}]_1 + 1; \sqrt{t}) + p_\mathcal{N}([\boldsymbol{x}]_1 - 1; \sqrt{t})\right)\prod_{i=2}^{d} p_\mathcal{N}([\boldsymbol{x}]_i; \sqrt{t})$, and the (vector-valued) ESF is given by $\nabla \log p_t^{(n)}(\boldsymbol{x}) = [\partial_1 \log p_t^{(n)}(\boldsymbol{x}), ..., \partial_d \log p_t^{(n)}(\boldsymbol{x})]$, where

$$\partial_1 \log p_t^{(n)}(\boldsymbol{x}) = \left(\hat{x}_t^{(n)}([\boldsymbol{x}]_1) - [\boldsymbol{x}]_1\right)/t\,,$$

$$\forall i \in \{2, ..., n\}\,, \quad \partial_i \log p_t^{(n)}(\boldsymbol{x}) = -[\boldsymbol{x}]_i/t\,, \tag{19}$$

where $\hat{x}_t^{(n)}$ is defined in the same way as in (7). Relative to the subspace to which the training set belongs — the $[\boldsymbol{x}]_1$-axis — we may refer to the first dimension as the *tangent* component and the other dimensions as the *normal* component. The tangent component of the ESF behaves the same as in the 1-D setting, whereas its normal components are linear functions and explain the uniform collapse onto the 1-D subspace during denoising.

To understand score smoothing in this context, we study an extension of the variational problem (11) to the multi-dimensional case through a generalization of the non-smoothness measure (10) by:

$$R^{(d)}[\boldsymbol{f}] := \sup_{(\boldsymbol{w},\boldsymbol{b})\in(\mathbb{S}^{d-1},\mathbb{R}^d)} \int_{-\infty}^{\infty} \|\nabla_{\boldsymbol{w}}^2 \boldsymbol{f}(\boldsymbol{w}x + \boldsymbol{b})\|dx \tag{20}$$

for $\boldsymbol{f}: \mathbb{R}^d \to \mathbb{R}^d$. Note that $R^{(d)}$ reduces to (10) when $d = 1$ and is invariant to coordinate rotations and translations.[3] We can now consider an analog of (11) for $d > 1$ by having (20) as the objective:

$$r_{t,\epsilon,d}^* := \inf_{\boldsymbol{f}} \quad R^{(d)}[\boldsymbol{f}] \qquad\qquad \text{s.t.} \qquad L_t^{(n)}[\boldsymbol{f}] < \epsilon\,, \tag{21}$$

with the infimum taken over all piece-wise twice-differentiable functions $\boldsymbol{f}: \mathbb{R}^d \to \mathbb{R}^d$.

Analogously to the 1-D case, we show that near-optimality can be achieved by vector-valued functions of the following type, $\hat{\boldsymbol{s}}_{t,\delta}^{(n)}: \mathbb{R}^d \to \mathbb{R}^d$, where $\hat{s}_{t,\delta}^{(n)}: \mathbb{R} \to \mathbb{R}$ is defined the same way as in (9):

$$\hat{\boldsymbol{s}}_{t,\delta}^{(n)}(\boldsymbol{x}) := [\hat{s}_{t,\delta}^{(n)}([\boldsymbol{x}]_1), -[\boldsymbol{x}]_2/t, ..., -[\boldsymbol{x}]_d/t]^\mathsf{T}\,. \tag{22}$$

(22) can be viewed as defining a generalization of the Smoothed PL-ESF to higher dimensions.

**Proposition 4** *Given any fixed $\epsilon \in (0, 0.015)$, if we choose $\delta_t = \kappa\sqrt{t}$ with any $\kappa \geq F^{-1}(\epsilon)$, then there exists $t_1 > 0$ (dependent on $\kappa$) such that the following holds for all $t \in (0, t_1)$:*

- $L_t^{(n)}[\hat{\boldsymbol{s}}_{t,\delta_t}^{(n)}] < \epsilon$;

- $R^{(d)}[\hat{\boldsymbol{s}}_{t,\delta_t}^{(n)}] < (1 + 8\sqrt{\epsilon})r_{t,\epsilon,d}^*$.

The proof is given in Appendix F and builds on Proposition 1 by leveraging two main observations: (1) $p_t^{(n)}$ is a product distribution between the tangent and normal dimensions; (2) the normal components of the ESF are fully linear and do not incur "additional" non-smoothness penalty under (20).

**Implication for denoising dynamics** Motivated by Proposition 4 and similar to in Section 4, we consider a denoising dynamics under the smoothed score given by $\frac{d}{dt}\mathbf{x}_t = -\frac{1}{2}\hat{\boldsymbol{s}}_{t,\delta_t}^{(n)}(\mathbf{x}_t)$, which is noticeably decoupled across the different dimensions:

$$\frac{d}{dt}[\mathbf{x}_t]_1 = -\frac{1}{2}\hat{s}_{t,\delta_t}^{(n)}([\mathbf{x}_t]_1)\,, \tag{23}$$

$$\forall i \in \{2, ..., d\}\,, \quad \frac{d}{dt}[\mathbf{x}_t]_i = \frac{1}{2}[\mathbf{x}_t]_i/t\,. \tag{24}$$

Based on our analysis in the $d = 1$ case, we know that this dynamics has the following solution:

**Proposition 5** *For $0 \leq s \leq t < 1/\kappa^2$, the solution of (23, 24) is given by $\mathbf{x}_s = \Phi_{s|t}(\mathbf{x}_t) := [\phi_{s|t}([\mathbf{x}_t]_1), \sqrt{s/t}[\mathbf{x}_t]_2, ..., \sqrt{s/t}[\mathbf{x}_t]_d]$ with $\phi_{s|t}$ defined as in (14, 15). Hence, if run backward-in-time with the marginal distribution of $\mathbf{x}_{t_0}$ being $\hat{p}_{t_0}^{(n,t_0)} \times p_\mathcal{N}(\,\cdot\,; \sqrt{t_0}) \times ... \times p_\mathcal{N}(\,\cdot\,; \sqrt{t_0})$, at $t \in [0, t_0)$, $\mathbf{x}_t$ has marginal distribution $\hat{p}_t^{(n,t_0)} \times p_\mathcal{N}(\,\cdot\,; \sqrt{t}) \times ... \times p_\mathcal{N}(\,\cdot\,; \sqrt{t})$, where $\hat{p}_t^{(n,t_0)}$ satisfies (16) when $t > 0$ and (17) when $t = 0$.*

---

[3]We note that our definition of $R^{(d)}$ differs from the complexity measure in function space associated with regularized two-layer ReLU NN on multi-dimensional inputs, which has a more involved definition via the Radon transform and fractional powers of Laplacians (Ongie et al., 2020).

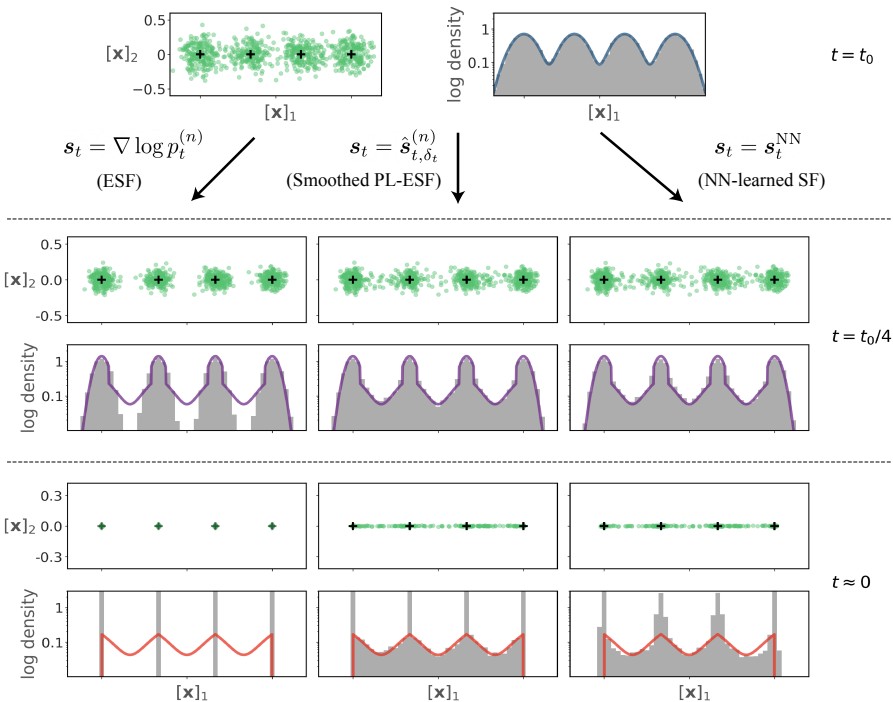

Figure 4: Results of the experiment in Section 6.1. Each column shows the denoising process under one of 3 choices of SFs, which starts from the distribution $p_{t_0}^{(n)}$ at $t_0$ and evolves backward-in-time following the respective SF. At $t = t_0$, $t_0/4$ and $t_{\min} = 10^{-5}$, we plot **(a)** the samples from the denoising processes in $\mathbb{R}^2$ and **(b)** the density histograms (log scale) of their first dimension. In **(b)**, the colored curves are the analytical predictions of $\hat{p}_t^{(n,t_0)}$ (for $t = t_0$, $t_0/4$) and $\tilde{p}_0^{(n,t_0)}$ (for $t = t_{\min}$), with the formulas given in Appendix C.2. A video animation of the three denoising processes and the evolution of the corresponding SFs can be found at this link.

Crucially, we see distinct dynamical behaviors in the tangent versus normal dimensions. As $t \to 0$, the trajectory converges to zero at a rate of $\sqrt{t}$ in the normal directions, resulting in a uniform collapse onto the $[\boldsymbol{x}]_1$-axis (as in the case *without* score smoothing). In the tangent dimension, meanwhile, a smoothing phenomenon happens similarly to the 1-D case. In particular, if the marginal distribution of $[\mathbf{x}_t]_1$ has a positive density on $[\delta_t - 1, 1 - \delta_t]$, then so will $[\mathbf{x}_0]_1$ on $[-1, 1]$, meaning that $\mathbf{x}_0$ has a non-singular density that interpolates smoothly among the training data on the desired 1-D subspace.

**Contrast with inference-time early stopping** The effect of score smoothing is different from what can be achieved by denoising under the exact ESF but stopping it at some $t_{\min} > 0$. In the latter case, the terminal distribution is still supported in all $d$ dimensions and is equivalent to simply corrupting the training data by Gaussian noise. Hence, without modifying the ESF, early stopping alone does not induce a proper generalization behavior.

## 6 NUMERICAL EXPERIMENTS

The detailed setup of all experiments is given in Appendix J.

### 6.1 DENOISING WITH SMOOTHED PL-ESF AND NN-LEARNED SF ($d = 2, n = 4$)

To validate our theoretical analysis on the effect of score smoothing on the denoising dynamics, we choose the setup in Section 5 with $d = 2$ and $n = 4$ and run the denoising dynamics (2) under three choices of the SF: **(i)** the ESF ($\boldsymbol{s}_t = \nabla \log p_t^{(n)}$), **(ii)** the Smoothed PL-ESF ($\boldsymbol{s}_t = \hat{\boldsymbol{s}}_{t,\delta_t}^{(n)}$ from

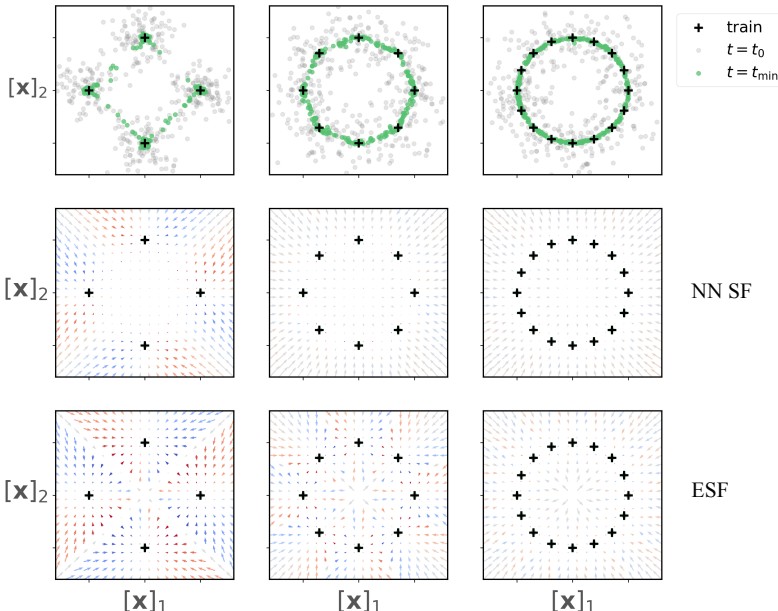

Figure 5: Experiment in Section 6.2 with training data spaced uniformly on the unit circle in $\mathbb{R}^2$. **Top**: Samples from the beginning and end of the denoising process with NN-learned SF. **Middle** and **bottom**: Visualization of the NN-learned SF vs ESF at $t = t_0/8$ as vector fields, with the length corresponding to the magnitude and the color determined by their angular direction (red for clockwise, blue for counter-clockwise).

(22)), and **(iii)** an NN-learned SF with $t$ as an input ($s_t = s_t^{\mathrm{NN}}$). All three processes are initialized at $t_0 = 0.02$ with the same marginal distribution $\mathbf{x}_{t_0} \sim p_{t_0}^{(n)}$ and run backward-in-time to $t_{\min} = 10^{-5}$.

The results are illustrated in Figure 4. We first observe that, in all three cases, the variance of the data distribution along the second dimension shrinks gradually to zero at a roughly similar rate as $t \to 0$, consistent with the argument in Section 5 that score smoothing does not interfere with the convergence in the normal direction. Meanwhile, in contrast with Col. **(i)**, where the variance along the first dimension shrinks to zero as well, we see in Col. **(ii)** that the variance along the first dimension remains positive for all $t$, validating the interpolation effect caused by smoothing the ESF. Moreover, the density histograms in Col. **(ii)** are closely matched by our analytical predictions of $\hat{p}_t^{(n,t_0)}$ and $\tilde{p}_0^{(n,t_0)}$ (the colored curves). Finally, we observe that Col. **(iii)** is much closer to **(ii)** than **(i)** in terms of how the distribution (as well as the SF itself, as shown in Figures 7 - 9) evolves during denoising. This suggests that NN learning causes a similar smoothing effect on the SF and supports the relevance of our theoretical analysis for understanding how NN-based DMs avoid memorization.

## 6.2 DATA ON A CIRCLE

To show that the effect of score smoothing goes beyond linearly-spaced data, we consider training sets spaced uniformly on a circle in $\mathbb{R}^2$ and train 2L NNs *without* weight decay to fit the ESF and drive the denoising dynamics, with results illustrated in Figure 5. As shown in the last two rows, the NN-learned SF is evidently smoother than the ESF, especially in the polar (i.e., tangent) direction. Notably, this leads the denoising dynamics to generate samples that interpolate between nearby training data points in a nearly linear fashion, which forms a *regular polygon* that approximates the underlying circle increasingly well as the number of training data increases. That this phenomenon occurs without weight decay in NN training also suggests that *implicit* regularization from gradient-based optimization can be sufficient to induce the score smoothing effect (see discussion in Appendix A).

In addition, we observe that NN-learned SF also tend to be smoother than the ESF when the training data are non-uniformly spaced in 1-D. The results are illustrated in Figure 10 in Appendix J.2.

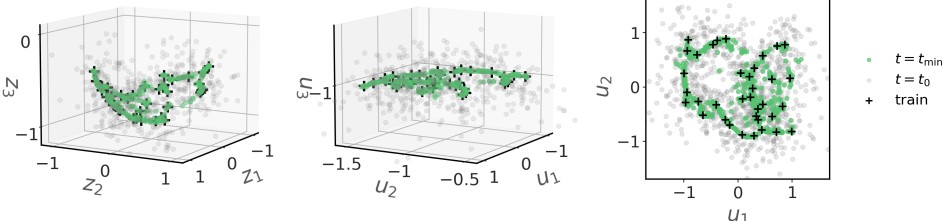

Figure 6: Results of the experiment in Section 6.3, where the training set is randomly sampled from a 2-D spherical-type manifold embedded in $d = 20$. Under different coordinate systems and views, the plots each illustrate a) the samples generated by denoising diffusion with the score function learned by a two-layer NN (green dots), b) their starting positions before denoising (gray dots), and c) the training set (black "plus" signs). *Left*: In the canonical 3-D subspace containing the manifold with coordinates $\boldsymbol{z} = [z_1, z_2, z_3]^\mathsf{T}$. *Right*: In latent coordinates produced by stereographic projection (Appendix J.4), i.e., $\boldsymbol{u} = [u_1, u_2, u_3]^\mathsf{T} := \Phi(\boldsymbol{z})$. *Bottom*: 2-D-sliced views in the $u_1$-$u_2$ plane.

## 6.3 CURVED MANIFOLDS IN HIGHER DIMENSIONS

In this section, we empirically demonstrate that the interpolation effect continues to exist when training data belong to manifolds in higher dimensions. A challenge for this goal is the difficulty to visualize interpolation phenomena in high dimensions. In light of this, we consider training data belonging to 2-D spherical-type manifolds embedded in higher dimensions (up to 20), which are amenable to visualization via stereographic projection onto a flattened latent space (Figure 11). Further details of the latent embedding and the experiment setup are given in Appendix J.4.

We consider training sets both randomly sampled and regularly spaced and train two- and three-layer NNs (without weight decay) for fitting the score function. The generated samples are visualized in Figures 6 and 12 - 14. We observe that NN-learned score estimators produce samples that interpolate the training data along the data manifold *despite* the nonlinearity and higher dimensionality of the data, suggesting that our analysis on the interpolation effect goes beyond linear subspace settings.

## 7 CONCLUSIONS AND LIMITATIONS

Through theoretical analyses and numerical experiments, our work shows how score smoothing can enable the denoising dynamics to produce distributions on the training data subspace without fully memorizing the training set. Further, by showing connections between NN learning and function smoothing, our results shed light on a general mechanism behind the generalization behavior of NN-based diffusion models via score smoothing. Additionally, as NN learning is just *one* way to achieve score smoothing, our work also motivates the exploration of alternative score estimators that facilitate generalization in DMs, such as the proposal by Scarvelis et al. (2025).

The present work focuses on a vastly simplified setup compared to real-world scenarios. It will be valuable to extend our theory to cases where training data are generally spaced, random or belonging to complex manifolds as well as to more general variants of DMs (De Bortoli et al., 2021; Albergo et al., 2025; Lipman et al., 2023; Liu et al., 2023). In addition, score estimators in practice typically have more complex architectures than 2L ReLU NN, which may exhibit more complex effects of regularization (Kamb and Ganguli, 2025). The connection between score smoothing and the *implicit bias* of NN training is also only explored to a limited extent, especially in the higher-dimensional setting (see Appendix A). Lastly, it will be useful to consider alternative forms of function smoothing and other regularization mechanisms beyond smoothing (Wibisono et al., 2024; Baptista et al., 2025).

**Reproducibility statement** Appendices D - H contain all the theoretical proofs; the experimental details can be found in Appendix J and at https://github.com/google-research/diffusion-score-smoothing.

**Acknowledgment.** The author thanks Zhengjiang Lin, Pengning Chao, Pranjal Awasthi, Arnaud Doucet, Eric Vanden-Eijnden, Binxu Wang and Molei Tao for valuable conversations and suggestions.

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

# A   RELATED WORKS

## A.1   MEMORIZATION AND GENERALIZATION IN DMs

Several works have noted the transition from generalization to memorization behaviors in DMs when the model capacity increases relatively to the training set size (Gu et al., 2025; Yi et al., 2023; Carlini et al., 2023; Kadkhodaie et al., 2024; Li et al., 2024b). Using tools from statistical physics, Biroli et al. (2024) showed that the transition to memorization occurs in the crucial regime where $t$ is small relative to the training set sparsity, which is also the focus of our study. Concurrent efforts by Ye et al. (2026); Zhang et al. (2025; 2026) also reveal that the generalization-to-memorization transition can be reflected in the size of the NN needed to represent the score function as well as the structure of the learned internal representation.

To derive rigorous learning guarantees, one line of work showed that DMs can produce a distribution accurately given a good score estimator (Song et al., 2021b; Lee et al., 2022; De Bortoli, 2022; Chen et al., 2023a;c; Shah et al., 2023; Cole and Lu, 2024; Benton et al., 2024; Huang et al., 2025), which leaves open the question of how to estimate the SF of an underlying density from finite training data without overfitting. For score estimation, when the ground truth density or its SF belongs to certain function classes, prior works have constructed score estimators with guaranteed sample complexity (Block et al., 2020; Li et al., 2023; Zhang et al., 2024; Wibisono et al., 2024; Chen et al., 2024; Gatmiry et al., 2024; Boffi et al., 2025; Han et al., 2024), including for scenarios where the data are supported on low-dimensional sub-manifolds (further discussed below). An end-to-end error bound is derived by Wang et al. (2024) which covers both training and sampling and is used to inform the choice of time and variance schedules. Unlike these results, which concern the estimation of densities from i.i.d. samples, our analysis does not assume a ground truth distribution. Based on a finite and fixed training set, our work focuses on the geometry of the SF when $t$ is small relative to the training set sparsity and elucidates how it determines the memorization behavior via an interplay with the denoising dynamics. For future work, it will be interesting to study the implication of score smoothing in the density estimation setting by potentially adapting our analysis to cases with randomly-sampled training data.

## A.2   DMs AND THE MANIFOLD HYPOTHESIS

An influential hypothesis is that high-dimensional real-world data often lie in low-dimensional sub-manifolds (Tenenbaum et al., 2000; Peyré, 2009), and it has been argued that DMs can estimate their intrinsic dimensions (Stanczuk et al., 2024; Kamkari et al., 2024), learn manifold features in meaningful orders (Wang and Vastola, 2023; 2024; Wang and Pehlevan, 2025; Achilli et al., 2024), or perform subspace clustering implicitly (Wang et al., 2025). Under the manifold hypothesis, Pidstrigach (2022); De Bortoli (2022); Potaptchik et al. (2025); Huang et al. (2024); Li and Yan (2024) have studied the convergence of DMs assuming a sufficiently good approximation to the true SF, while Oko et al. (2023); Chen et al. (2023b); Azangulov et al. (2024) have proved sample complexity guarantees for score estimation using NN models. In particular, prior works such as Chen et al. (2023b); Wang and Vastola (2024); Gao and Li (2024); Ventura et al. (2025) have considered the decomposition of the SF into tangent and normal components. Our work is novel in showing how score smoothing can affect these two components differently: reducing the speed of convergence towards training data along the *tangent* direction (to avoid memorization) while preserving it along the *normal* direction (to ensure a convergence onto the subspace).

## A.3   SCORE SMOOTHING AND REGULARIZATION

Aithal et al. (2024) showed empirically that NNs tend to learn smoother versions of the ESF and argued that this leads to a mode interpolation effect that explains model hallucination. Scarvelis et al. (2025) designed alternative closed-form DMs by smoothing the ESF, although the theoretical analysis therein is limited to showing that their smoothed SF is directed towards certain barycenters of the training data. Their work inspired our further theoretical analysis on how score smoothing affects the denoising dynamics and leads to a terminal distribution that interpolates the training data. In the context of image generation, Kamb and Ganguli (2025) showed that imposing locality and equivariance to the score estimator allows the model to generalize better. However, in addition to being limited to the image generation setting with CNN-based score estimators, their result does not

show *how* new samples are generated, but only that *if* such samples are created, then they obey the consistency properties enforced by the CNN architecture. In comparison, our work shows that the interpolation effect can arise from score smoothing with NN architectures as simple as 2L MLPs.

Recent works including Wibisono et al. (2024); Baptista et al. (2025) considered other SF regularizers such as the empirical Bayes regularization (capping the magnitude in regions where $p_t^{(n)}$ is small) or Tikhonov regularization (constraining the norm averaged over $p_t^{(n)}$). In the linear subspace setting, these methods tend to reduce the magnitude of the SF in not only the tangent but also the normal directions, thus slowing down the convergence onto the subspace and resulting in a terminal distribution that still has a $d$-dimensional support. In contrast, the Smoothed PL-ESF preserves the (linear) normal component and hence is not prone to this issue.

Building on a convex program formulation of the training of two-layer ReLU-like NNs in finite-data settings, Zhang and Pilanci (2025) derive a convex program relevant to score learning with two-layer NNs. Compared to the score matching loss (4), however, the optimization objective therein goes through an additional simplification step: the expectation over the noised empirical distribution is approximated by sampling one noise vector per training data point. This choice crucially reduces the problem into a finite-data setting in order to derive the convex program formulations, but it introduces variance to the problem as a result. Furthermore, whereas Zhang and Pilanci (2025) prove that the score estimator obtained by solving the convex program generates Gaussian or Gauss-Laplace distributions through Langevin Monte Carlo, we show that a smoothed score drives the denoising dynamics to produce distributions on desired subspaces that interpolate the training data.

When the training data either are orthogonal or form a so-called "obtuse-simplex", a concurrent work by Zeno et al. (2025) showed that DMs equipped with SFs learned by minimal-cost two-layer NNs can generate data points near the boundary of the corresponding hyperbox / simplex. Our work focuses instead on the 1-D-data-subspace setting (hence not orthogonal but parallel) and proves how score smoothing induced by NN regularization causes the denoising dynamics to generate samples that interpolate the training data along the subspace. In terms of the theoretical characterization, their work makes the simplifying assumption that the NN perfectly fits the ESF (only) on non-intersecting balls centered at the training data points – i.e., approximating the true noised empirical distribution by a compact distribution in the definition of the score matching loss (4) – thus bypassing part of the complexity that arises from the interplay between the full score matching loss and the non-smoothness penalty in the variational problems (11) and (21).

### A.4   (EXPLICIT OR IMPLICIT) NN REGULARIZATION AS NON-SMOOTHNESS PENALTY

Savarese et al. (2019) showed that the weight norm of an infinite-width two-layer ReLU NN on scalar inputs is equivalent to a complexity measure in function space given by (10). This result motivates us to define and study the variational problem (11) which is specific to score learning: minimizing the non-smoothness $R[f]$ subject to the constraint of the score matching loss $L^{(n)}[f]$. Our novel theoretical contribution on this front is the analysis of this variational problem, in particular proving that a near-minimizer of this problem at small $t$ is given by the Smoothed PL-ESF (Proposition 1). This required a novel analysis of the interplay among the score matching objective, the smoothness penalty, and the geometry of the ESF. The result from Savarese et al. (2019) was later extended by Ongie et al. (2020) to the case of multi-dimensional inputs, and it will be very interesting to adapt our analysis in the multi-dimensional case to this complexity measure.

Though models in practice are not always trained with explicit regularization such as weight decay, a similar effect could still occur through *implicit* regularization: it has been shown theoretically that when trained with gradient-based algorithms for sufficiently long, certain classes of NNs can be viewed as implicitly minimizing some complexity measure while fitting the target labels (Soudry et al., 2018; Ji and Telgarsky, 2019; Lyu and Li, 2020). Notably, when we consider infinite-width 2-layer homogeneous NNs trained with logistic-type losses (and under certain assumptions), the complexity measure agrees with the one in (10) (Chizat and Bach, 2020). Though the argument does not apply directly in our setting as score estimation involves a different type of loss, it gives us reason to hypothesize that score smoothing can occur through NN training even without explicit regularization. It would be highly relevant to investigate this further in future work. In addition, a concurrent work by Bonnaire et al. (2025) also considers score estimation with random feature

networks and shows how implicit regularization from its gradient flow training dynamics can give rise to the model's generalization behavior.

# B  ADDITIONAL NOTATIONS

We use big-O notations only for denoting asymptotic relations as $t \to 0$. Specifically, for functions $f, g : \mathbb{R}_+ \to \mathbb{R}_+$, we will write $f(t) = O(g(t))$ if $\exists t_1, C > 0$ (they may depend on other variables such as $\kappa$ and $\Delta$) such that $\forall t \in (0, t_1)$, it holds that $f(t) \le Cg(t)$. In addition, in several situations where $f$ decays exponentially fast in $1/t$ as $t \to 0$ but the exact exponent is not of much importance, we will simply write $f(t) = O(\exp(-C/t))$, which is intended to be interpreted as $\exists C > 0$ such that $f(t) = O(\exp(-C/t))$ (and the value of $C$ can differ in different contexts).

# C  GENERALIZATION TO $n > 2$

The analysis above can be generalized to the scenario where $S$ consists of $n > 2$ points spaced uniformly on an interval $[-D, D]$, that is, $y_k := 2(k-1)\Delta - D$ for $k \in [n]$, where $\Delta := D/(n-1) = (y_{k+1} - y_k)/2$. We additionally define $z_k := y_k + \Delta = (y_k + y_{k+1})/2$ for $k \in [n-1]$.

## C.1  SCORE SMOOTHING

In this case, we can still express the ESF as (6) except for replacing (7) by

$$\hat{x}_t^{(n)}(x) := \frac{\sum_{k=1}^n y_k p_{\mathcal{N}}(x - y_k, \sqrt{t})}{\sum_{k=1}^n p_{\mathcal{N}}(x - y_k, \sqrt{t})} \ , \tag{25}$$

and its PL approximation at small $t$ is now given by

$$\bar{s}_t^{(n)}(x) := \begin{cases} (y_1 - x)/t \ , & \text{if } x \le z_1 \ , \\ (y_k - x)/t \ , & \text{if } x \in [z_{k-1}, z_k] \text{ for } k \in \{2, ..., n-1\} \ , \\ (y_n - x)/t \ , & \text{if } x \ge z_{n-1} \ . \end{cases} \tag{26}$$

For $\delta \in (0, \Delta)$, we now define

$$\hat{s}_{t,\delta}^{(n)}(x) := \begin{cases} (y_1 - x)/t \ , & \text{if } x \le y_1 + \delta \ , \\ (y_n - x)/t \ , & \text{if } x \ge y_{n-1} - \delta \ , \\ (y_k - x)/t \ , & \text{if } x \in [y_k - \delta, y_k + \delta], \exists k \in [n] \ , \\ \delta/(\Delta - \delta) \cdot (x - z_k)/t \ , & \text{if } x \in [y_{k-1} + \delta, y_k - \delta], \exists k \in [n-1] \ , \end{cases} \tag{27}$$

and it is not hard to show that Proposition 2 and Lemma 9 can be generalized to the following results, with their proofs given in Appendix E.1 and E, respectively:

**Proposition 6** *Let $\delta_t = \kappa\sqrt{t}$ for some $\kappa > 0$. Then $\exists t_1, C > 0$ such that $\forall t \in (0, t_1)$,*

$$\tfrac{n-1}{n} F(\kappa) - C\sqrt{t} \le L_t^{(n)}[\hat{s}_{t,\delta_t}^{(n)}] \le \tfrac{n-1}{n} F(\kappa) + C\sqrt{t} \ , \tag{28}$$

*where $t_1$ and $C$ depend only on $\kappa$ and $F$ is a function that strictly decreases from $1$ to $0$ on $[0, \infty)$.*

**Lemma 7** *$\exists t_1, C > 0$ such that $\forall t \in (0, t_1)$, it holds that*

$$t \cdot \mathbb{E}_{x \sim p_t^{(n)}} \left[ \left\| \bar{s}_t^{(n)}(x) - (\tfrac{d}{dx} \log p_t^{(n)})(x) \right\|^2 \right] \le C \exp(-\delta^2/(9t)) \ . \tag{29}$$

## C.2  DENOISING DYNAMICS

The backward-in-time dynamics of (13) can also be solved analytically in a similar fashion, where (14) is replaced by

$$\phi_{s|t}(x) := \begin{cases} \sqrt{\tfrac{s}{t}}(x - y_1) + y_1 \ , & \text{if } x \le y_1 + \delta_t \ , \\ \sqrt{\tfrac{s}{t}}(x - y_n) + y_n \ , & \text{if } x \ge y_n - \delta_t \ , \\ \sqrt{\tfrac{s}{t}}(x - y_k) + y_k \ , & \text{if } x \in [y_k - \delta_t, y_k + \delta_t], \exists k \in \{2, ..., n-1\} \ , \\ \tfrac{\Delta - \delta_s}{\Delta - \delta_t}(x - z_k) + z_k \ , & \text{if } x \in [y_k + \delta_t, y_{k+1} - \delta_t], \exists k \in [n-1] \ . \end{cases} \tag{30}$$

The formula (16) is then generalized to

$$
\hat{p}_s^{(n,t_0)}(x) = \begin{cases}
\delta_t/\delta_s \cdot \hat{p}_t^{(n,t_0)}(\delta_t/\delta_s \cdot x - (\delta_t - \delta_s)/\delta_s \cdot y_1) \,, & \text{if } x \leq y_1 + \delta_s \\
\delta_t/\delta_s \cdot \hat{p}_t^{(n,t_0)}(\delta_t/\delta_s \cdot x - (\delta_t - \delta_s)/\delta_s \cdot y_n) \,, & \text{if } x \geq y_n - \delta_s \\
\delta_t/\delta_s \cdot \hat{p}_t^{(n,t_0)}(\delta_t/\delta_s \cdot x - (\delta_t - \delta_s)/\delta_s \cdot y_k) \,, & \text{if } x \in [y_k - \delta_s, y_k + \delta_s] \\
(\Delta - \delta_t)/(\Delta - \delta_s) \cdot \hat{p}_t^{(n,t_0)}((\Delta - \delta_t)/(\Delta - \delta_s) \cdot x + (\delta_t - \delta_s)/(\Delta - \delta_s) \cdot z_k) \,, \\
\qquad\qquad\qquad\qquad\qquad\qquad\qquad\qquad\qquad\qquad \text{if } x \in [y_k + \delta_s, y_{k+1} - \delta_s]
\end{cases}
\tag{31}
$$

When $s = 0$, there is

$$
\phi_{0|t}(x) = \begin{cases}
(\Delta x - z_k \delta_t)/(\Delta - \delta_t) \,, & \text{if } x \in [y_k + \delta_t, y_{k+1} - \delta_t], \exists k \in [n-1] \,, \\
y_{\arg\min_k |y_k - x|} \,, & \text{otherwise.}
\end{cases}
\tag{32}
$$

As $\phi_{0|t}(x)$ is invertible when restricted to $\cup_{k\in[n-1]}[y_k + \delta_t, y_{k+1} - \delta_t]$, the terminal distribution can be decomposed as

$$
\hat{p}_0^{(n,t_0)} = \sum_{k=1}^{n} a_k \boldsymbol{\delta}_{y_k} + \left(1 - \sum_{k=1}^{n} a_k \boldsymbol{\delta}_{y_k}\right)\tilde{p}_0^{(n,t_0)} \,,
\tag{33}
$$

where $\tilde{p}_0^{(n,t_0)}$ is a probability distribution defined as

$$
\tilde{p}_0^{(n,t_0)}(x) = \begin{cases}
(\Delta - \delta_t)/\Delta \cdot \hat{p}_t^{(n,t_0)}((\Delta - \delta_t)/\Delta \cdot x + \delta_t/\Delta \cdot z_k) \,, & \text{if } x \in [y_k, y_{k+1}] \\
0 \,, & \text{otherwise} \,.
\end{cases}
\tag{34}
$$

and it holds for *all* $t \in (0, t_0]$ that

$$
a_k = \begin{cases}
\mathbb{E}_{\mathbf{x}\sim\hat{p}_t^{(n,t_0)}}\left[\mathbb{1}_{\mathbf{x}\geq -D+\delta_t}\right] \,, & \text{if } k = 1 \,, \\
\mathbb{E}_{\mathbf{x}\sim\hat{p}_t^{(n,t_0)}}\left[\mathbb{1}_{\mathbf{x}\geq D-\delta_t}\right] \,, & \text{if } k = n \,, \\
\mathbb{E}_{\mathbf{x}\sim\hat{p}_t^{(n,t_0)}}\left[\mathbb{1}_{y_k-\delta_t\leq \mathbf{x}\leq y_k+\delta_t}\right] \,, & \text{if } k \in \{2, ..., n-1\} \,.
\end{cases}
\tag{35}
$$

## C.3 Higher Dimensions

Because the definition of $R^{(d)}$ is invariant to rotations and translations of the coordinate system, we can assume without loss of generality that the training set lies on the $[\boldsymbol{x}]_1$-axis and is spaced uniformly on the interval $[-D, D]$.

Thanks to the decoupling across dimensions, the denoising dynamics associated with the generalized Smoothed PL-ESF also follows (23) and (24). Hence, Proposition 5 still holds except with (14) - (17) and (16) replaced by (30) - (32) and (34), respectively.

## D Proof of Proposition 1

The first claim is a straightforward consequence of Proposition 6: when $t$ is small enough (with threshold dependent on $\kappa$), there is

$$
L_t^{(n)}[\hat{s}_{t,\delta_t}^{(n)}] \leq \frac{n-1}{n}F(\kappa) + C\sqrt{t} < F(\kappa) \leq F(F^{-1}(\epsilon)) = \epsilon \,.
\tag{36}
$$

Next we consider the second claim. On one hand, it is easy to compute that

$$
R[\hat{s}_{t,\delta_t}^{(n)}] = \sum_{k=1}^{n-1} 2\left(\frac{\delta_t}{t(\Delta - \delta_t)} + \frac{1}{t}\right) = 2(n-1)\frac{\Delta}{t(\Delta - \delta_t)}
\tag{37}
$$

On the other hand, let $f$ be any function on $\mathbb{R}$ that belongs to the feasible set of the minimization problem (11), meaning that $f$ is twice differentiable except on a set of measure zero and $L_t^{(n)}[f] < \epsilon$. Define $\epsilon_k := t\int_{-\infty}^{\infty}|f(x) - \frac{d}{dx}\log p_t^{(n)}(x)|^2 p_{\mathcal{N}}(x - y_k; \sqrt{t})dx$ for each $k \in [n]$. By the definition

of $p_t^{(n)}$, we then have $\sum_{k=1}^n \epsilon_k < n\epsilon$. If we consider a change-of-variable $\tilde{x} = (x - y_k)/\sqrt{t}$ and define $\tilde{f}_k(\tilde{x}) := \sqrt{t} f(y_k + \sqrt{t}\tilde{x})$, there is

$$
\int_{y_k-\Delta}^{y_k+\Delta} |f(x) - \bar{s}_t^{(n)}(x)|^2 p_{\mathcal{N}}(x - y_k; \sqrt{t}) dx = \int_{y_k-\Delta}^{y_k+\Delta} \left| f(x) - \frac{y_k - x}{t} \right|^2 p_{\mathcal{N}}(x - y_k; \sqrt{t}) dx
$$
$$
= t^{-1} \int_{-\Delta/\sqrt{t}}^{\Delta/\sqrt{t}} |\tilde{f}_k(\tilde{x}) + \tilde{x}|^2 p_{\mathcal{N}}(\tilde{x}; 1) d\tilde{x} .
$$
(38)

Hence, using (62) with $\delta = \Delta$, we obtain that

$$
\int_{-\Delta/\sqrt{t}}^{\Delta/\sqrt{t}} |\tilde{f}_k(\tilde{x}) + \tilde{x}|^2 p_{\mathcal{N}}(\tilde{x}; 1) d\tilde{x} \le t \int_{-\infty}^{\infty} |f(x) - \bar{s}_t^{(n)}(x)|^2 p_{\mathcal{N}}(x - y_k; \sqrt{t}) dx
$$
$$
\le t \int_{-\infty}^{\infty} |f(x) - \tfrac{d}{dx} \log p_t^{(n)}(x)|^2 p_{\mathcal{N}}(x - y_k; \sqrt{t}) dx
$$
(39)
$$
+ t \int_{-\infty}^{\infty} \left| \tfrac{d}{dx} \log p_t^{(n)}(x) - \bar{s}_t^{(n)}(x) \right|^2 p_{\mathcal{N}}(x - y_k; \sqrt{t}) dx
$$
$$
\le \epsilon_k + O(\exp(-\Delta^2/(9t))) .
$$

Thus, for $t$ small enough such that $\sqrt{t} < \Delta/3$, we can apply Lemma 8 from below to $\tilde{f}_k$, from which we obtain (after reversing the change-of-variable) that

$$
\inf_{x \in [y_k - 1.5\sqrt{t}, y_k + 1.5\sqrt{t}] \setminus N} f'(x) \le (-1 + 2\sqrt{\epsilon_k})/t + O(\exp(-\Delta^2/(10t)))
$$
$$
\inf_{x \in [y_k + 1.5\sqrt{t}, y_k + 3\sqrt{t}]} f(x) \le -0.5
$$
(40)
$$
\sup_{x \in [y_k - 3\sqrt{t}, y_k - 1.5\sqrt{t}]} f(x) \ge 0.5 .
$$

Hence, $\exists a_k \in [y_k - 1.5\sqrt{t}, y_k + 1.5\sqrt{t}] \setminus N$, $b_{k,+} \in [y_k + 1.5\sqrt{t}, y_k + 3\sqrt{t}]$ and $b_{k,-} \in [y_k - 3\sqrt{t}, y_k - 1.5\sqrt{t}]$ such that $f'(a_k) \le (-1 + 2\sqrt{\epsilon_k})/t + O(\exp(-\Delta^2/(10t)))$, $f(b_{k,+}) \le 0$ and $f(b_{k,-}) \ge 0$. Furthermore, for $t$ small enough such that $\sqrt{t} < \Delta/6$, there is $b_{k,+} < b_{k+1,-}$ for $k \in [n-1]$, and hence by the fundamental theorem of calculus, $\exists c_k \in [b_{k,+}, b_{k+1,-}]$ such that $f'(c_k) \ge 0$.

Now, we focus on the sequence of points, $a_1 < c_1 < a_2 < ... < c_{n-1} < a_n$. By the fundamental theorem of calculus and the fact that $f$ is twice differentiable except for on a finite set, there is

$$
\int_{a_k}^{c_k} |f''(x)| dx \ge |f'(c_k) - f'(a_k)| \ge (1 - 2\sqrt{\epsilon_k})/t - O(\exp(-\Delta^2/(10t)))
$$
$$
\int_{c_k}^{a_{k+1}} |f''(x)| dx \ge |f'(a_{k+1}) - f'(c_k)| \ge (1 - 2\sqrt{\epsilon_{k+1}})/t - O(\exp(-\Delta^2/(10t)))
$$

and hence it is clear that

$$
R[f] = \int_{-\infty}^{\infty} |f''(x)| dx \ge \sum_{k=1}^{n-1} |f'(c_k) - f'(a_k)| + \sum_{k=1}^{n-1} |f'(a_{k+1}) - f'(c_k)|
$$
(41)
$$
\ge 2\Big(n - 1 - \sum_{k=1}^{n} 2\sqrt{\epsilon_k}\Big)/t - O(\exp(-\Delta^2/(10t)))
$$
$$
\ge 2(n - 1 - 2n\sqrt{\epsilon})/t - O(\exp(-\Delta^2/(10t))) .
$$

Therefore, for $0 < \epsilon < 0.015$, as $n \ge 2$, it holds for $t$ sufficiently small that

$$
\frac{R[\hat{s}_{t,\delta_t}^{(n)}]}{R[f]} \le \frac{(n-1)\Delta}{(n - 1 - 2n\sqrt{\epsilon})(\Delta - \delta_t) - O(\exp(-\Delta^2/(10t)))}
$$
(42)
$$
\le 1 + 7.9\sqrt{\epsilon} + O(\sqrt{t}) ,
$$

which is bounded by $1 + 8\sqrt{\epsilon}$. Since this holds for any $f$ in the feasible set, it also holds when the denominator on the left-hand-side is replaced by the infimum, $r_{t,\epsilon}^*$.

$\square$

**Lemma 8** *Suppose $f$ is twice differentiable on $\mathbb{R}$ except on a set $N$ of measure zero and $\int_{-3}^{3} |x + f(x)|^2 p_{\mathcal{N}}(x;1)dx < \epsilon$ with $0 < \epsilon < 0.03$. Then we have*

$$\inf_{x \in [-1.5, \, 1.5] \setminus N} f'(x) \leq -1 + 2\sqrt{\epsilon} \tag{43}$$

$$\inf_{x \in [1.5, \, 3]} f(x) \leq -0.5 \tag{44}$$

$$\sup_{x \in [-1.5, \, -3]} f(x) \geq 0.5 \tag{45}$$

*Proof of Lemma 8*: We first prove (43) by supposing for contradiction that $\inf_{x \in [-1.5, \, 1.5] \setminus N} f'(x) = -1 + k_\Delta$ with $k_\Delta > 2\sqrt{\epsilon}$. By the fundamental theorem of calculus, this means that the function $x + f(x)$ is monotonically increasing with slope at least $k_\Delta$ on $[-1.5, 1.5]$. Hence, there exists $x_1 \in \mathbb{R}$ such that $|x + f(x)| > k_\Delta |x - x_1|$ for $x \in [-1.5, 1.5]$. Therefore,

$$
\begin{aligned}
\int_{-\infty}^{\infty} |x + f(x)|^2 p_{\mathcal{N}}(x;1)dx &\geq \int_{-1.5}^{1.5} |x + f(x)|^2 p_{\mathcal{N}}(x;1)dx \\
&\geq (k_\Delta)^2 \int_{-1.5}^{1.5} |x - x_1|^2 p_{\mathcal{N}}(x;1)dx \\
&= (k_\Delta)^2 \int_{-1.5}^{1.5} (x^2 + 2xx_1 + (x_1)^2) p_{\mathcal{N}}(x;1)dx \\
&\geq (k_\Delta)^2 \int_{-1.5}^{1.5} x^2 p_{\mathcal{N}}(x;1)dx \\
&> 0.25(k_\Delta)^2 > \epsilon \,,
\end{aligned}
\tag{46}
$$

which shows a contradiction.

Next, we prove (44) by supposing for contradiction that $\inf_{x \in [1.5, \, 3] \setminus N} f(x) > -0.5$, in which case it holds that $\inf_{1.5 \leq x \leq 3} |f(x) + x| > 1$, and hence

$$
\begin{aligned}
\int_{-\infty}^{\infty} |x + f(x)|^2 p_{\mathcal{N}}(x;1)dx &\geq \int_{1.5}^{3} |x + f(x)|^2 p_{\mathcal{N}}(x;1)dx \\
&\geq \int_{1.5}^{3} p_{\mathcal{N}}(x;1)dx \\
&> 0.03 > \epsilon
\end{aligned}
\tag{47}
$$

which shows a contradiction. A similar argument can be used to prove (45).

$\square$

## E  PROOF OF PROPOSITION 2

Below we prove Proposition 6, which generalizes Proposition 2 to the case where $n > 2$.

**Lemma 9** $\exists t_1, C > 0$ *such that* $\forall t \in (0, t_1)$*, it holds that*

$$t \cdot \mathbb{E}_{x \sim p_t^{(n)}} \left[ \left\| \bar{s}_t^{(n)}(x) - (\tfrac{d}{dx} \log p_t^{(n)})(x) \right\|^2 \right] \leq C \exp(-\delta^2/(9t)) \,. \tag{48}$$

Lemma 9 is proved in Appendix E.1. In light of it, we only need to show that

$$t \int_{-\infty}^{\infty} |\hat{s}_{t,\delta_t}^{(n)}(x) - \bar{s}_t^{(n)}(x)|^2 p_t^{(n)}(x)dx = \frac{n-1}{n} F(\kappa) + O(\sqrt{t}) \,. \tag{49}$$

By the definition of $p_t^{(n)}$, we can first evaluate the integral with respect to the density $p_\mathcal{N}(x - y_k; \sqrt{t})$ for each $k \in [n]$ separately and then sum them up. We define

$$y_{k,-} = \begin{cases} -\infty, & \text{if } k = 1 \\ y_k - \delta_t, & \text{otherwise} \end{cases}, \qquad y_{k,+} = \begin{cases} \infty, & \text{if } k = n \\ y_k + \delta_t, & \text{otherwise} \end{cases} \qquad (50)$$

By construction, $\hat{s}_{t,\delta_t}^{(n)}$ is a PL function whose slope is changed only at each $y_{k,-}$ and $y_{k,+}$.

Let us fix a $k \in [n]$. Since $\hat{s}_{t,\delta_t}^{(n)}(x) = \bar{s}_t^{(n)}(x)$ when $x \in [y_{k,-}, y_{k,+}]$, we only need to estimate the difference between the two functions outside of $[y_{k,-}, y_{k,+}]$.

We first consider the interval $(y_{k,+}, y_k + \Delta] = (y_k + \delta_t, y_k + \Delta]$ when $k \in \{1, ..., n-1\}$, on which it holds that

$$\hat{s}_{t,\delta_t}^{(n)}(x) - \bar{s}_t^{(n)}(x) = \frac{\Delta}{t} \cdot \frac{x - (y_k + \delta_t)}{\Delta - \delta_t}, \qquad (51)$$

by the piecewise-linearity of the two functions. Hence,

$$
\begin{aligned}
&t \int_{y_k + \delta_t}^{y_k + \Delta} |\hat{s}_{t,\delta_t}^{(n)}(x) - \bar{s}_t^{(n)}(x)|^2 p_\mathcal{N}(x - y_k; \sqrt{t}) dx \\
&= \left( \frac{\Delta}{\Delta - \delta_t} \right)^2 \int_{y_k + \delta_t}^{y_k + \Delta} \left| \frac{x - y_k}{\sqrt{t}} - \frac{\delta_t}{\sqrt{t}} \right|^2 p_\mathcal{N}(x - y_k; \sqrt{t}) dx \\
&= \left( \frac{\Delta}{\Delta - \delta_t} \right)^2 \left( \int_{y_k + \delta_t}^{\infty} \left| \frac{x - y_k}{\sqrt{t}} - \kappa \right|^2 p_\mathcal{N}(x - y_k; \sqrt{t}) dx \right. \\
&\qquad\qquad\qquad \left. - \int_{y_k + \Delta}^{\infty} \left| \frac{x - y_k}{\sqrt{t}} - \kappa \right|^2 p_\mathcal{N}(x - y_k; \sqrt{t}) dx \right)
\end{aligned} \qquad (52)
$$

Note that by a change-of-variable $\tilde{x} \leftarrow (x - y_k)/\sqrt{t}$, we obtain that

$$\int_{y_k + \delta_t}^{\infty} \left| \frac{x - y_k}{\sqrt{t}} - \kappa \right|^2 p_\mathcal{N}(x - y_k; \sqrt{t}) dx = \frac{1}{2} F(\kappa), \qquad (53)$$

where we define

$$F(\kappa) := 2 \int_\kappa^\infty |u - \kappa|^2 p_\mathcal{N}(u; 1) du \qquad (54)$$

It is straightforward to see that, as $\kappa$ increases from $0$ to $\infty$, $F$ strictly decreases from $1$ to $0$. Therefore,

$$
\begin{aligned}
&t \int_{y_k + \delta_t}^{y_k + \Delta} |\hat{s}_{t,\delta_t}^{(n)}(x) - \bar{s}_t^{(n)}(x)|^2 p_\mathcal{N}(x - y_k; \sqrt{t}) dx \\
&= \left( \frac{\Delta}{\Delta - \delta_t} \right)^2 \left( \frac{1}{2} F(\kappa) - \int_{\Delta/\sqrt{t}}^\infty |u - \kappa|^2 p_\mathcal{N}(u; 1) dx \right) \\
&= \frac{1}{2} F(\kappa) + O(\sqrt{t})
\end{aligned} \qquad (55)
$$

Next, we consider the interval $[y_k + \Delta, \infty)$, in which we have

$$|\hat{s}_{t,\delta_t}^{(n)}(x) - \bar{s}_t^{(n)}(x)| \le \frac{\Delta}{t}. \qquad (56)$$

Thus,

$$
\begin{aligned}
t \int_{y_k + \Delta}^\infty |\hat{s}_{t,\delta_t}^{(n)}(x) - \bar{s}_t^{(n)}(x)|^2 p_\mathcal{N}(x - y_k; \sqrt{t}) dx &\le t \int_{y_k + \Delta}^\infty \left| \frac{\Delta}{t} \right|^2 p_\mathcal{N}(x - y_k; \sqrt{t}) dx \\
&= \frac{\Delta^2}{t} \int_{\Delta/\sqrt{t}}^\infty p_\mathcal{N}(u; 1) du \\
&= O\left( t^{-1} \exp\left( -\frac{\Delta^2}{2t} \right) \right)
\end{aligned} \qquad (57)
$$

Hence, we have

$$t \int_{y_k+\delta_t}^{\infty} |\hat{s}_{t,\delta_t}^{(n)}(x) - \bar{s}_t^{(n)}(x)|^2 p_{\mathcal{N}}(x - y_k; \sqrt{t})dx = \frac{1}{2}F(\kappa) + O(\sqrt{t}) \,. \tag{58}$$

Similarly, for $k \in \{2, ..., n\}$, we can show that

$$t \int_{-\infty}^{y_k-\delta_t} |\hat{s}_{t,\delta_t}^{(n)}(x) - \bar{s}_t^{(n)}(x)|^2 p_{\mathcal{N}}(x - y_k; \sqrt{t})dx = \frac{1}{2}F(\kappa) + O(\sqrt{t}) \,. \tag{59}$$

Thus, there is

$$
\begin{aligned}
&t \int_{-\infty}^{\infty} |\hat{s}_{t,\delta_t}^{(n)}(x) - \bar{s}_t^{(n)}(x)|^2 p_{\mathcal{N}}(x - y_k; \sqrt{t})dx \\
&= \begin{cases} F(\kappa) + O(\sqrt{t}) \,, & \text{if } k \in \{2, ..., n-1\} \\ \frac{1}{2}F(\kappa) + O(\sqrt{t}) \,, & \text{if } k = 1 \text{ or } n \,. \end{cases}
\end{aligned}
\tag{60}
$$

Summing them together, we get that

$$t \int_{-\infty}^{\infty} |\hat{s}_{t,\delta_t}^{(n)}(x) - \bar{s}_t^{(n)}(x)|^2 p_t^{(n)}(x)dx = \frac{n-1}{n}F(\kappa) + O(\sqrt{t}) \,. \tag{61}$$

This proves the proposition.

$\square$

### E.1   PROOF OF LEMMA 9

Below we prove Lemma 7, which generalizes Lemma 9 to the case where $n > 2$.

By the definition of $p_t^{(n)}$, it suffices to show that $\forall k \in [n]$,

$$\int |\tfrac{d}{dx} \log p_t^{(n)}(x) - \bar{s}_t^{(n)}(x)|^2 p_{\mathcal{N}}(x - y_k; \sqrt{t})dx = O(\exp(-\delta^2/(9t))) \,. \tag{62}$$

Consider any $k \in [n]$. We decompose the integral into three intervals and bound them separately. First, when $x \geq y_k + \frac{1}{2}\Delta > y_k$, noticing that $|\tfrac{d}{dx} \log p_t^{(n)}(x)|, |\bar{s}_t^{(n)}(x)| \leq (|x| + 2D)/t$, we obtain that

$$
\begin{aligned}
&\int_{y_k+\frac{1}{2}\Delta}^{\infty} \left| \tfrac{d}{dx} \log p_t^{(n)}(x') - \bar{s}_t^{(n)}(x') \right|^2 p_{\mathcal{N}}(x - y_k; \sqrt{t})dx' \\
&\leq \frac{1}{\sqrt{2\pi t}} \int_{y_k+\frac{1}{2}\Delta}^{\infty} \left| \tfrac{d}{dx} \log p_t^{(n)}(x') - \bar{s}_t^{(n)}(x') \right|^2 \exp\left( -\frac{(x'-y_k)^2}{2t} \right) dx' \\
&\leq \frac{1}{\sqrt{2\pi t}} \int_{y_k+\frac{1}{2}\Delta}^{\infty} \frac{4(|x'| + 2D)^2}{t^2} \exp\left( -\frac{(x'-y_k)^2}{2t} \right) dx' \\
&\leq \frac{1}{\sqrt{2\pi t}} \int_{y_k+\frac{1}{2}\Delta}^{\infty} \frac{4(|x'-y_k| + 4D)^2}{t^2} \exp\left( -\frac{(x'-y_k)^2}{2t} \right) dx' \\
&\leq \frac{8}{\sqrt{2\pi t^3}} \int_{y_k+\frac{1}{2}\Delta}^{\infty} \left( \left(\frac{x'-y_k}{\sqrt{t}}\right)^2 + \frac{16D^2}{t} \right) \exp\left( -\frac{(x'-y_k)^2}{2t} \right) dx' \\
&= \frac{8}{\sqrt{2\pi t^3}} \int_{\frac{\delta}{2\sqrt{t}}}^{\infty} \left( (\tilde{x})^2 + \frac{16D^2}{t} \right) \exp\left( -\frac{(\tilde{x})^2}{2} \right) d\tilde{x} \\
&\leq \frac{8}{\sqrt{2\pi t^3}} \left( \left(\frac{16D^2}{t} + 1\right)\sqrt{\frac{\pi}{2}} + \frac{\delta}{2\sqrt{t}} \right) \exp\left( -\frac{\delta^2}{8t} \right) = O\left( \exp\left( -\frac{\delta^2}{9t} \right) \right) \,,
\end{aligned}
\tag{63}
$$

where for the last inequality we use Lemma 10 below.

A similar bound can be derived when the range of the outer integral is changed to between $-\infty$ and $y_k - \frac{1}{2}\Delta$.

Next, suppose $x \in [y_k - \frac{1}{2}\Delta, y_k + \frac{1}{2}\Delta]$, which means that $|x - y_k| \leq \frac{1}{2}\Delta$ while $|x - y_l| \geq \frac{3}{2}\Delta$ for $l \neq k$. Thus, it holds for any $l \neq k$ that

$$\frac{p_{\mathcal{N}}(x - y_k; \sqrt{t})}{p_{\mathcal{N}}(x - x_l; \sqrt{t})} = \exp\left(-\frac{|x - y_k|^2 - |x - x_l|^2}{2t}\right) \geq \exp\left(\frac{\Delta\delta}{t}\right) \tag{64}$$

Hence, writing $q_{t,k}(x) := \frac{p_{\mathcal{N}}(x - y_k; \sqrt{t})}{\sum_{l=1}^{n} p_{\mathcal{N}}(x - x_l; \sqrt{t})}$, there is $q_{t,k}(x) \geq 1 - (n-1)\exp\left(-\frac{\Delta\delta}{t}\right)$ and for $l \neq k$, $q_{t,l}(x) < \exp\left(-\frac{\Delta\delta}{t}\right)$. Therefore,

$$\begin{aligned}
\left|\frac{d}{dx}\log p_t^{(n)}(x) - \bar{s}_t^{(n)}(x)\right| &\leq \frac{|(q_{t,k}(x) - 1)y_k| + \sum_{l \neq k}|q_{t,k}(x)y_k|}{t} \\
&\leq \frac{2(n-1)D}{t}\exp\left(-\frac{\Delta\delta}{t}\right) = O\left(t^{-1}\exp\left(-\frac{\Delta\delta}{t}\right)\right).
\end{aligned} \tag{65}$$

Since $p_{\mathcal{N}}(x - y_k; \sqrt{t}) \leq \frac{1}{\sqrt{2\pi t}}$ for any $x'$, we then have

$$\begin{aligned}
&\int_{y_k - \frac{1}{2}\Delta}^{y_k + \frac{1}{2}\Delta} \left|\frac{d}{dx}\log p_t^{(n)}(x') - \bar{s}_t^{(n)}(x')\right|^2 p_{\mathcal{N}}(x - y_k; \sqrt{t})dx' \\
&\leq \frac{\Delta + w}{\sqrt{2\pi t}} \sup_{y_k - \frac{1}{2}\Delta \leq x' \leq y_k + \frac{1}{2}\Delta} \left|\frac{d}{dx}\log p_t^{(n)}(x') - \bar{s}_t^{(n)}(x')\right|^2 \\
&= O\left(t^{-3}\exp\left(-\frac{2\Delta\delta}{t}\right)\right)
\end{aligned} \tag{66}$$

Combining (63) with (66) yields the desired result.

$\square$

**Lemma 10** *For $u \geq 0$,*

$$\int_u^{\infty} e^{-x^2/2}dx \leq \sqrt{\frac{\pi}{2}}e^{-u^2/2} \tag{67}$$

$$\int_u^{\infty} x^2 e^{-x^2/2}dx \leq \left(\sqrt{\frac{\pi}{2}} + u\right)e^{-u^2/2} \tag{68}$$

*Proof of Lemma 10*: It known (e.g., Chang et al. 2011) that

$$\int_u^{\infty} e^{-x^2}dx \leq \frac{\sqrt{\pi}}{2}e^{-u^2}, \tag{69}$$

from which (67) can be obtained by a simple change-of-variable.

Next, using integration-by-parts, we obtain that

$$\begin{aligned}
\int_a^b x^2 e^{-x^2/2}dx &= x(-e^{-x^2/2})\big|_a^b - \int_a^b 1 \cdot (-e^{-\frac{x^2}{2}})dx \\
&= (ae^{-a^2/2} - be^{-b^2/2}) + \int_a^b e^{-x^2/2}dx
\end{aligned} \tag{70}$$

Hence,

$$\begin{aligned}
\int_u^{\infty} x^2 e^{-x^2/2}dx &\leq \int_u^{\infty} e^{-x^2/2}dx + ue^{-u^2/2} \\
&\leq \left(\sqrt{\frac{\pi}{2}} + u\right)e^{-u^2/2}
\end{aligned} \tag{71}$$

$\square$

## F  PROOF OF PROPOSITION 4

To make the dependence on $d$ more explicit, below we add it as a super-script into $p_t^{(n,d)}$ and $L_t^{(n,d)}$.

Our first observation is that the score matching loss $L_t^{(n,d)}[\boldsymbol{f}]$ can be decomposed as a sum over components that depend separately the different output dimensions of $\boldsymbol{f}$: $L_t^{(n,d)}[\boldsymbol{f}] = \sum_{i=1}^{d} L_{t,i}^{(n,d)}[[\boldsymbol{f}]_i]$ with $L_{t,i}^{(n,d)}[f] := t \cdot \mathbb{E}_{\mathbf{x} \sim p_t^{(n)}}[\|f(\mathbf{x}) - \partial_i \log p_t^{(n)}(\mathbf{x})\|^2]$ . In particular, in each of the normal dimensions, $L_{t,i}^{(n,d)}$ can achieve its global minimum of 0 if we set $[\boldsymbol{f}(\boldsymbol{x})]_i = \partial_i \log p_t^{(n)}(\boldsymbol{x}) = -[\boldsymbol{x}]_i/t$ for $i > 1$. This motivates us to focus on candidate solutions of the following form:

**Definition 11** *Given $g : \mathbb{R} \to \mathbb{R}$, we define a **flat extension** of $g$ along the normal directions to the domain $\mathbb{R}^d$, $f_g : \mathbb{R}^d \to \mathbb{R}$, by*

$$f_g(\boldsymbol{x}) := g([\boldsymbol{x}]_1) , \ \forall \boldsymbol{x} \in \mathbb{R}^d . \tag{72}$$

*We then further define*

$$\boldsymbol{f}_g(\boldsymbol{x}) := [f_g(\boldsymbol{x}), -[\boldsymbol{x}]_2/t, ..., -[\boldsymbol{x}]_d/t] . \tag{73}$$

**Lemma 12** *Given any $g : \mathbb{R} \to \mathbb{R}$, it holds that $L_t^{(n,d)}[\boldsymbol{f}_g] = L_t^{(n,1)}[g]$ and $R^{(d)}[\boldsymbol{f}_g] = R^{(1)}[g]$.*

*Proof of Lemma 12:* For the first property, we note that

$$
\begin{aligned}
L_{t,1}^{(n,d)}[f_g] &= t \cdot \mathbb{E}_{\mathbf{x} \sim p_t^{(n,d)}}[\|f_g(\mathbf{x}) - \partial_1 \log p_t^{(n,d)}(\mathbf{x})\|^2] \\
&= t \cdot \mathbb{E}_{\mathbf{x} \sim p_t^{(n,d)}}\left[\left\|g([\mathbf{x}]_1) - \frac{d}{d[\mathbf{x}]_1} \log p_t^{(n,1)}([\mathbf{x}]_1)\right\|^2\right] \\
&= t \cdot \mathbb{E}_{\mathbf{x} \sim p_t^{(n,1)}}\left[\left\|g(\mathbf{x}) - \frac{d}{d\mathbf{x}} \log p_t^{(n,1)}(\mathbf{x})\right\|^2\right] \\
&= L_t^{(n,1)}[g] .
\end{aligned}
\tag{74}
$$

Hence, $L_t^{(n,d)}[\boldsymbol{f}_g] = L_{t,1}^{(n,d)}[\boldsymbol{f}_g] = L_t^{(n,1)}[g]$.

For the second property, let us consider any $\boldsymbol{w} \in \mathbb{S}^{d-1}$. There is

$$\nabla_{\boldsymbol{w}} f_g(\boldsymbol{x}) = \boldsymbol{w} \cdot \nabla f_g(\boldsymbol{x}) = [\boldsymbol{w}]_1 g'([\boldsymbol{x}]_1) , \tag{75}$$

$$\nabla_{\boldsymbol{w}}^2 f_g(\boldsymbol{x}) = [\boldsymbol{w}]_1 \partial_{\boldsymbol{w}} g'([\boldsymbol{x}]_1) = ([\boldsymbol{w}]_1)^2 g''([\boldsymbol{x}]_1) . \tag{76}$$

Meanwhile, thanks to the linearity in the normal dimensions, we have $[\nabla_{\boldsymbol{w}}^2 \boldsymbol{f}_g(\boldsymbol{x})]_i \equiv 0, \ \forall i \in \{2, ..., d\}$. Therefore,

$$
\begin{aligned}
\int_{-\infty}^{\infty} \|\nabla_{\boldsymbol{w}}^2 \boldsymbol{f}_g(\boldsymbol{w}x + \boldsymbol{b})\| dx &= \int_{-\infty}^{\infty} |\nabla_{\boldsymbol{w}}^2 f_g(\boldsymbol{w}x + \boldsymbol{b})| dx \\
&= ([\boldsymbol{w}]_1)^2 \int_{-\infty}^{\infty} |g''([\boldsymbol{w}]_1 x + [\boldsymbol{b}]_1)| dx \\
&= |[\boldsymbol{w}]_1| R^{(1)}[g] \le R^{(1)}[g] .
\end{aligned}
\tag{77}
$$

Since we maximize over all $\boldsymbol{w} \in \mathbb{S}^{d-1}$ in the definition of $R_1^{(d)}$, we see that $R^{(d)}[\boldsymbol{f}_g] = R^{(1)}[g]$.

$\square$

**Lemma 13** $r_{t,\epsilon,d}^* \ge r_{t,\epsilon,1}^*, \ \forall d \in \mathbb{N}_+^d.$

*Proof of Lemma 13:* Consider any $\tilde{\boldsymbol{f}} : \mathbb{R}^d \to \mathbb{R}^d$ in the feasible set (i.e., $L_t^{(n,d)}[\tilde{\boldsymbol{f}}] < \epsilon$), and we let $\tilde{f} = [\tilde{\boldsymbol{f}}]_1$ denote its tangent dimension. This implies that $L_{t,1}^{(n,d)}[\tilde{f}] \le L_t^{(n,d)}[\tilde{\boldsymbol{f}}] < \epsilon$. On the other hand, we have

$$L_{t,1}^{(n,d)}[\tilde{f}] = t \cdot \mathbb{E}_{\mathbf{x}_2,...,\mathbf{x}_d \sim p_{\mathcal{N}}(0;\sqrt{t})}\left[\mathbb{E}_{\mathbf{x}_1 \sim p_t^{(n,1)}}\left[\left|\tilde{f}(\mathbf{x}_1, \mathbf{x}_2, ..., \mathbf{x}_d) - \frac{d}{d\mathbf{x}_1} \log p_t^{(n,1)}(\mathbf{x}_1)\right|^2\right]\right] \ge \tilde{l} , \tag{78}$$

where we let $\tilde{l} := \inf_{(x_2,...,x_d) \in \mathbb{R}^{d-1}} \mathbb{E}_{\mathrm{x} \sim p_t^{(n,1)}}[|\tilde{f}(\mathrm{x}, x_2, ..., x_d) - \frac{d}{d\mathrm{x}} \log p_t^{(n,1)}(\mathrm{x})|^2]$. For any given $\epsilon_0 > 0$, we can find $x_2^*, ..., x_d^* \in \mathbb{R}^{d-1}$ such that

$$\mathbb{E}_{\mathrm{x} \sim p_t^{(n,1)}} \left[ \left| \tilde{f}(\mathrm{x}, x_2^*, ..., x_d^*) - \frac{d}{d\mathrm{x}} \log p_t^{(n,1)}(\mathrm{x}) \right|^2 \right] < \tilde{l} + \epsilon_0 \tag{79}$$

We now define $\tilde{g} : \mathbb{R} \to \mathbb{R}$ by

$$\tilde{g}(x) := \tilde{f}(x, x_2^*, ..., x_d^*) . \tag{80}$$

Then (79) implies that $L_t^{(n,1)}[\tilde{g}] < \tilde{l} + \epsilon_0 \le L_{t,1}^{(n,d)}[\tilde{f}] + \epsilon_0$. Since $\epsilon_0$ is arbitrary, we may choose $\epsilon_0 = \epsilon - L_{t,1}^{(n,d)}[\tilde{f}]$, which then implies $L_t^{(n,1)}[\tilde{g}] < \epsilon$, and hence $\tilde{g}$ is in the feasible set for the 1-D variational problem. On the other hand, we see that

$$\begin{aligned} R^{(1)}[\tilde{g}] &= \int_{-\infty}^{\infty} \left| \frac{d^2}{dx^2} \tilde{f}(x, x_2^*, ..., x_d^*) \right| dx \\ &= \int_{-\infty}^{\infty} \left| \nabla_{\boldsymbol{w}}^2 \tilde{f}(\boldsymbol{w}x + \boldsymbol{b}) \right| dx \le \int_{-\infty}^{\infty} \left\| \nabla_{\boldsymbol{w}}^2 \tilde{\boldsymbol{f}}(\boldsymbol{w}x + \boldsymbol{b}) \right\| dx , \end{aligned} \tag{81}$$

if we choose $\boldsymbol{w} = [1, 0, ..., 0]$ and $\boldsymbol{b} = [0, x_2^*, ..., x_d^*]$. This implies that $R^{(1)}[\tilde{g}] \le R^{(d)}[\tilde{\boldsymbol{f}}]$.

Since this argument applies to any $\tilde{\boldsymbol{f}}$ in the feasible set, we can take infimum over $\tilde{\boldsymbol{f}}$ to conclude that $r_{t,\epsilon,d}^* \ge r_{t,\epsilon,1}^*$.

$\square$

Now we proceed to the proof of the proposition. Let $\epsilon$ and $\kappa$ be specified as in the proposition statement, and let $t_1$ be determined in the same way as in Proposition 1. We observe that $[\hat{s}_{t,\delta_t}^{(n)}]_1$ is the *flat extension* of $s_{t,\delta_t}^{(n)}$ to the domain $\mathbb{R}^d$, and therefore, $\hat{s}_{t,\delta_t}^{(n)} = \boldsymbol{f}_{s_{t,\delta_t}^{(n)}}$. By Lemma 12, this means that $L_t^{(n,d)}[\hat{s}_{t,\delta_t}^{(n)}] = L_t^{(n,1)}[s_{t,\delta_t}^{(n)}]$, and the right-hand-side is smaller than $\epsilon$ when $t < t_1$ by Proposition 1. Meanwhile, Lemma 12 also implies that $R^{(d)}[\hat{s}_{t,\delta_t}^{(n)}] = R^{(1)}[s_{t,\delta_t}^{(n)}]$, and the right-hand-side is smaller than $(1 + 8\sqrt{\epsilon})r_{t,\epsilon,1}^*$ when $t < t_1$ Proposition 1, which is further bounded above by $(1 + 8\sqrt{\epsilon})r_{t,\epsilon,d}^*$ by Lemma 13. This completes the proof of Proposition 4.

## G  PROOF OF PROPOSITION 3

We consider each of three cases separately.

**Case I:** $x \in [\delta_t - 1, 1 - \delta_t]$.   In this case, it is easy to verify that $x_s = \frac{1 - \delta_s}{1 - \delta_t} x$ is a valid solution to the ODE

$$\frac{d}{ds} x_s = -\frac{1}{2} \frac{\delta_s}{1 - \delta_s} \frac{x_s}{s} , \tag{82}$$

on $[0, t]$ that satisfies the terminal condition $x_t = x$. It remains to verify that for all $s \in (0, t)$, it holds that $x_s \in [\delta_s - 1, 1 - \delta_s]$ (i.e., the entire trajectory during $[0, t]$ remains in region **C**).

Suppose that $x \ge 0$. Then it is clear that $x_s \ge 0, \forall s \in [0, t]$. Moreover, it holds that

$$x_s - (1 - \delta_s) = \frac{1 - \delta_s}{1 - \delta_t}(x_t - (1 - \delta_t)) \le 0 \tag{83}$$

Therefore, $x_s \in [0, 1 - \delta_s] \subseteq [\delta_s - 1, 1 - \delta_s]$. A similar argument can be made if $x < 0$.

**Case II:** $x \le \delta_t - 1$.   In this case, it is also easy to verify that $x_s = \sqrt{\frac{s}{t}}(x + 1) - 1$ is a valid solution to the ODE

$$\frac{d}{ds} x_s = \frac{1}{2} \frac{x + 1}{s} , \tag{84}$$

on $[0, t]$ that satisfies the terminal condition $x_t = x$. It remains to verify that for all $s \in (0, t)$, it holds that $x_s \le \delta_s - 1$ (i.e., the entire trajectory during $[0, t]$ remains in region **A**). This is true because

$$(x_s + 1) - \delta_s = \sqrt{\frac{s}{t}}(x + 1) - \delta_s = \sqrt{\frac{s}{t}}(x + 1 - \delta_t) \le 0 . \tag{85}$$

**Case III:** $x \geq 1 - \delta_t$. A similar argument can be made as in Case II above.

$\square$

# H PROOF OF PROPOSITION 15

The proof relies on the following lemma, which allows us to relate the KL-divergence between $\hat{p}_0^{(n,t_0)}$ and the uniform density via that of $\hat{p}_{t_0}^{(n,t_0)}$:

**Lemma 14** $\forall t \in [0, t_0]$, $KL(u_1||\hat{p}_0^{(n,t_0)}) = KL(u_{1-\delta_t}||\hat{p}_t^{(n,t_0)})$.

*Proof of Lemma 14*:

$$
\begin{aligned}
\mathrm{KL}(u_1||\hat{p}_0^{(n,t_0)}) &= \int_{-1}^{1} \frac{1}{2} \cdot (-\log 2 - \log((1 - a_+ - a_-)\tilde{p}_0^{(n,t_0)}(x))) dx \\
&= -\log 2 - \frac{1}{2} \int_{-1}^{1} \log((1 - a_+ - a_-)\tilde{p}_0^{(n,t_0)}(x)) dx \\
&= -\log 2 - \frac{1}{2(1 - \delta_t)} \int_{\delta_t - 1}^{1 - \delta_t} \left( \log(1 - \delta_t) + \log(\hat{p}_t^{(n,t_0)}(x')) \right) dx' \\
&= \frac{1}{2(1 - \delta_t)} \int_{\delta_t - 1}^{1 - \delta_t} \log(1/(2(1 - \delta_t))) - \log(\hat{p}_t^{(n,t_0)}(x')) dx' \\
&= \mathrm{KL}(u_{1-\delta_t}||\hat{p}_t^{(n,t_0)})
\end{aligned}
\tag{86}
$$

$\square$

In light of Lemma 14, we choose $t = t_0$ and examine the KL-divergence between $u_{1-\delta_{t_0}}$ and $\hat{p}_{t_0}^{(n,t_0)}$. By symmetry, we only need to consider the right half of the interval, $[0, 1 - \delta_{t_0}]$, on which there is $\hat{p}_{t_0}^{(n,t_0)}(x) = p_{t_0}^{(n)}(x) \geq \frac{1}{2}p_{\mathcal{N}}(x - 1; \sqrt{t_0})$. We have

$$
\begin{aligned}
\int_{0}^{1 - \delta_{t_0}} \log\left(p_{\mathcal{N}}(x - 1; \sqrt{t_0})\right) dx &= \int_{-1}^{-\delta_{t_0}} \log\left(\frac{1}{\sqrt{2\pi t_0}} \exp(-x^2/t_0)\right) dx \\
&= -\frac{1 - \delta_{t_0}}{2}(\log(2\pi) + \log(t_0)) - \frac{1}{t_0} \int_{-1}^{-\delta_{t_0}} x^2 dx \\
&\geq -\frac{1 - \delta_{t_0}}{2}(\log(2\pi) + \log(t_0)) - \frac{1}{3t_0} .
\end{aligned}
\tag{87}
$$

Therefore,

$$
\begin{aligned}
\mathrm{KL}(u_{[0,1-\delta_{t_0}]}||\hat{p}_{t_0}^{(n,t_0)}) &= \frac{1}{1 - \delta_{t_0}} \int_{0}^{1 - \delta_{t_0}} -\log(1 - \delta_{t_0}) - \log\left(\frac{1}{2}p_{\mathcal{N}}(x - 1; \sqrt{t_0})\right) dx \\
&\leq -\log(1 - \delta_t) + \log(2) - \frac{1}{1 - \delta_{t_0}}\left(-\frac{1 - \delta_{t_0}}{2}(\log(2\pi) + \log(t_0)) - \frac{1}{3t_0}\right) \\
&\leq \frac{1}{3t_0(1 - \delta_{t_0})} + \log\left(\frac{\sqrt{t_0}}{1 - \delta_{t_0}}\right) + \log(2\sqrt{2\pi}) .
\end{aligned}
\tag{88}
$$

By symmetry, the same bound can be obtained for $\mathrm{KL}(u_{1-\delta_{t_0}}||\hat{p}_{t_0}^{(n,t_0)})$, which yields the desired result when combined with Lemma 14.

# I ADDITIONAL DISCUSSIONS

## I.1 CONNECTION TO LOCAL SMOOTHING

Given $f : \mathbb{R} \to \mathbb{R}$ and $w > 0$, we define a *locally-smoothed* version of $f$ with window size $w$ as:

$$
(w * f)(x) := \frac{1}{2w} \int_{x-w}^{x+w} f(x') dx' .
\tag{89}
$$

It is then not hard to see that, for $w \in (0, 1]$,

$$(w * \bar{s}_t^{(n)})(x) = \hat{s}_{t,1-w}^{(n)}(x), \quad \forall x \in \mathbb{R}. \tag{90}$$

Therefore, $\hat{s}_{t,\delta}^{(n)}$ can also be interpreted as a locally-smoothed version of the PL ESF with window size $1 - \delta$, where a smaller value of $\delta$ means a higher window size for local smoothing.

### I.2 KL DIVERGENCE BOUND

(18) allows us to prove KL-divergence bounds for $\hat{p}_0^{(n,t_0)}$ based on those of $\hat{p}_t^{(n,t_0)}$. For example, letting $u_a$ denote the uniform density on $[-a, a]$, we have:

**Proposition 15** *Let $\kappa > 0$ and $0 < t_0 < 1/\kappa^2$. If $\mathbf{x}_t$ solves (13) backward-in-time with $\mathbf{x}_{t_0} \sim p_{t_0}^{(n)}$, then there is $KL(u_1 || \hat{p}_0^{(n,t_0)}) \leq \frac{1}{3t_0(1-\kappa\sqrt{t_0})} + \log\left(\frac{\sqrt{t_0}}{1-\kappa\sqrt{t_0}}\right) + \log(2\sqrt{2\pi}) < \infty.$*

This result is proved in Appendix H. Although the choice of the uniform density as the target to compare $\hat{p}_0^{(n,t_0)}$ with is an arbitrary one (since the training set is fixed rather than sampled from the uniform distribution), the result still rigorously establishes the smooth component of $\hat{p}_0^{(n,t_0)}$ that interpolates the training set. In contrast, denoising with the exact ESF results in $p_0^{(n)}$, which is fully singular and has an infinite KL-divergence with *any* smooth density on $[-1, 1]$.

### I.3 COMPARISON WITH INFERENCE-TIME EARLY STOPPING

The effect of score smoothing on the denoising dynamics is different from what can be achieved by denoising under the exact ESF but stopping it at some positive $t_{\min}$. In the latter case, the terminal distribution is still supported in all $d$ dimensions and equivalent to corrupting the training data directly by Gaussian noise. In other words, without modifying the ESF, early stopping alone is not sufficient for inducing a proper generalization behavior.

## J ADDITIONAL DETAILS ON THE NUMERICAL EXPERIMENTS

All experiments were run on a hosted Jupyter Notebook service with a single TPU (v3) as backend. The code was written in JAX (Bradbury et al., 2018).

### J.1 NN-LEARNED SF VS SMOOTHED PL-ESF IN 1-D

**NN-learned SF.** We trained a two-layer MLP with hidden-layer-width 1024 and a skip linear connection from the input layer to fit the ESF at $t = 0.05$. The model is trained by the AdamW optimizer (Kingma and Ba, 2015; Loshchilov and Hutter, 2019) for 80000 steps with learning rate 0.0003 and various choices of the weight decay coefficient. At each training step, the optimization objective is an approximation of the expectation in (3) using a batch of 1024 i.i.d. samples from $p_t^{(n)}$.

**Smoothed PL-ESF.** We chose $t = 0.05$ and four values of $\delta$ ($\delta_1 = 0.648$, $\delta_2 = 0.548$, $\delta_3 = 0.453$, $\delta_4 = 0.346$), which were tuned to roughly match the corresponding curves in the left panel.

### J.2 EXPERIMENT OF SECTION 6.1

The ESF is computed from its analytical expression (25). To ensure numerical stability at small $t$, we truncate the sampled values of $\nabla \log p_t^{(n)}$ based on magnitude. At $t_0 = 0.02$, 20000 realizations of $\mathbf{x}_{t_0}$ are sampled from $p_{t_0}^{(n)}$. Then, the ODEs are numerically solved backward-in-time to $t = 10^{-5}$ using Euler's method under the noise schedule from Karras et al. (2022) with 200 steps and $\rho = 2$.

For the Smoothed PL-ESF, we choose $\delta_t = \kappa\sqrt{t}$ with $\kappa = 1.2$.

For the NN-learned SF, after a rescaling by $\sqrt{t}$ (c.f. the discussion on output scaling in Karras et al. 2022), we parameterize $\mathbf{s}_t^{\mathrm{NN}}$ with three two-layer MLP blocks ($\mathrm{MLP}_1$, $\mathrm{MLP}_2$, $\mathrm{MLP}_3$): $\mathrm{MLP}_1$

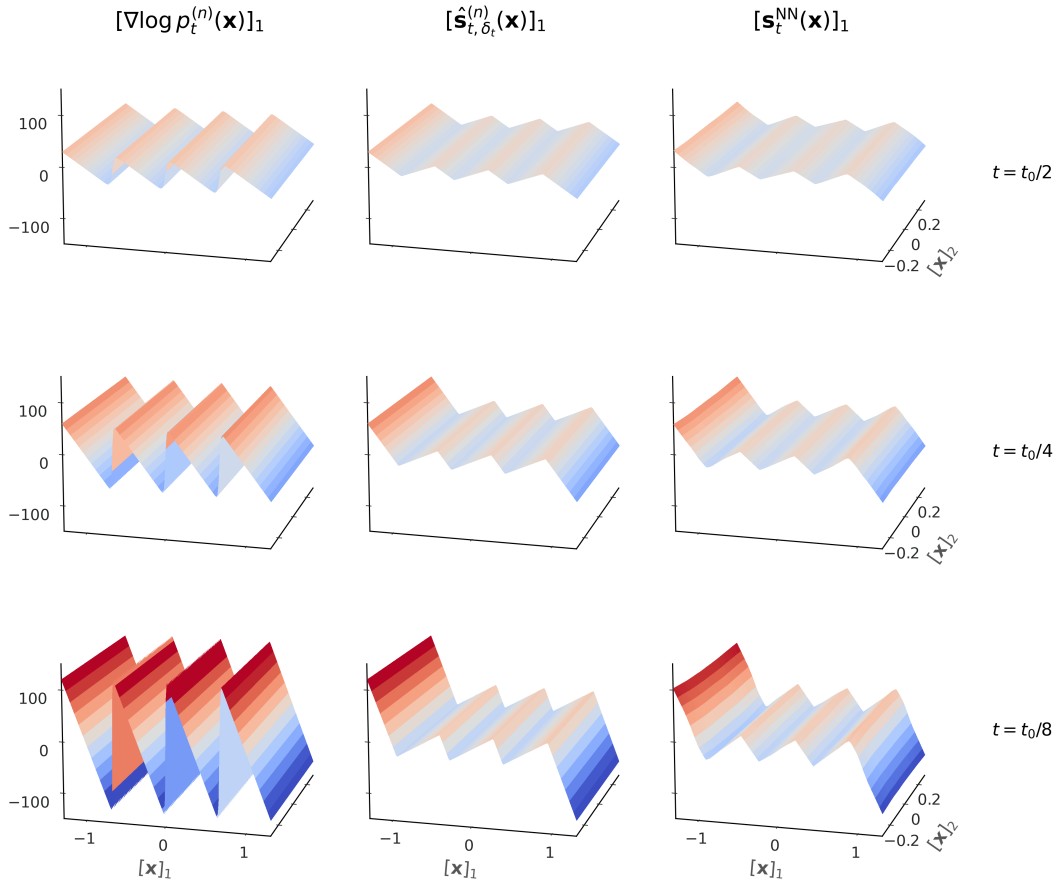

Figure 7: Comparing three SF variants from Experiment 2 in their first (tangent) dimension at different $t$. We see a close proximity between the Smoothed PL-ESF and the NN-learned SF, both of which are smoother than the ESF especially at small $t$.

is applied to $\log(t)$ to compute a time embedding; $\mathrm{MLP}_2$ is applied to the concatenation of $\boldsymbol{x}$ and the time embedding; $\mathrm{MLP}_3$ is also applied to $\log(t)$ and its output modulates the output of $\mathrm{MLP}_2$ similarly to the Adaptive Rescaling module (Perez et al., 2018; Peebles and Xie, 2023). $\mathrm{MLP}_1$ and $\mathrm{MLP}_3$ share the first-layer weights and biases. The model is trained to minimize a discretized version of (3) with $T = t_0$, where the integral is approximated by sampling $t$ from $[10^{-6}, t_0]$ with $t^{1/3}$ uniformly distributed (inspired by the noise schedule of Karras et al. 2022) and then $\boldsymbol{x}$ from $p_t^{(n)}$. The parameters are updated by the AdamW optimizer with learning rate 0.00005, batch size 1024 and a total number of 150000 steps, where weight decay (with coefficient 2) is applied only to the weights and biases of $\mathrm{MLP}_2$.

## J.3 EXPERIMENTS OF SECTION 6.2

**Data on a circle** The training data lies on a circle with radius 1 centered at the origin in $\mathbb{R}^2$, and we choose $t_0 = 0.08$ and $n = 4, 8$ and 16. The NN model is a two-layer MLP with hidden-layer-width 512 and it is trained with learning rate 0.0001, batch size 64 and no weight decay for 5000, 20000 and 80000 epochs respectively for the three choices of $n$. The rest of the configuration is the same as described in Appendix J.2.

**Randomly-spaced in 1-D** The training data (with $n = 6$) are randomly perturbed from a uniform grid. We choose $t = 0.1$. The NN model has the same architecture as described in Appendix J.1 and is trained for 15000 steps using the AdamW optimizer with learning rate 0.00005.

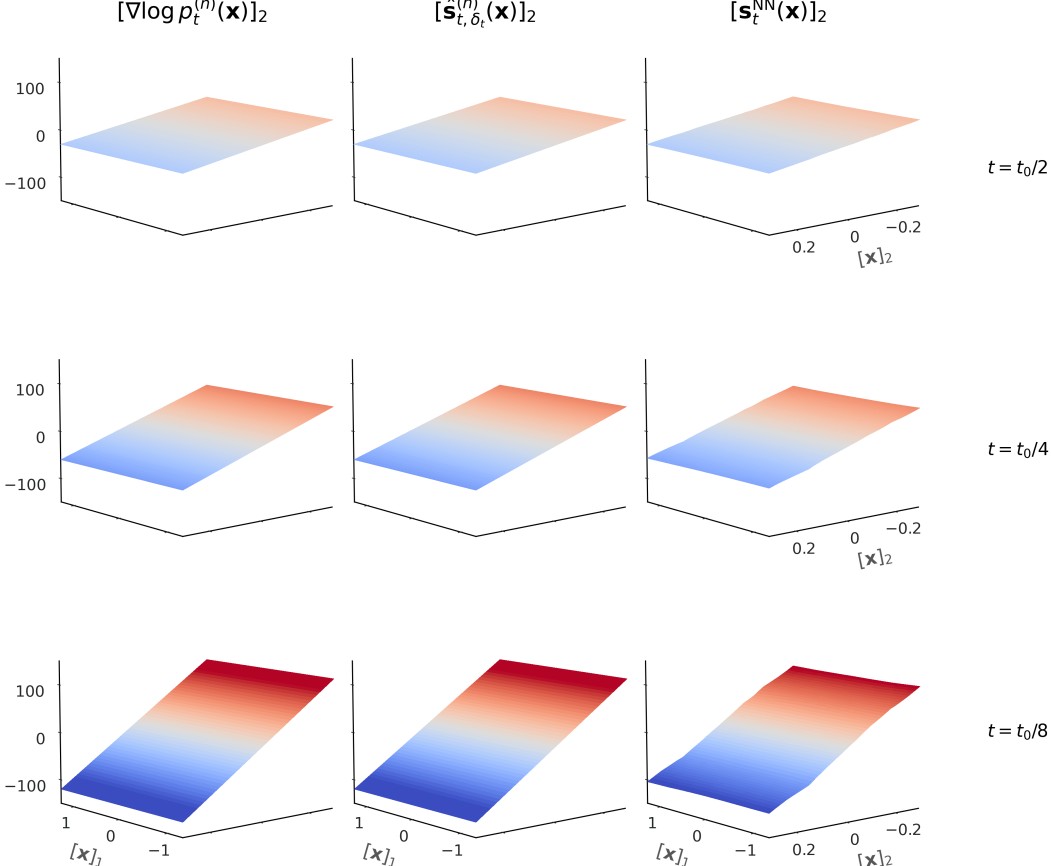

Figure 8: Comparing three SF variants from Experiment 2 in their second (normal) dimension at different $t$. We see all three SF are relatively similar in the normal direction, except for a mild distortion of the NN-learned SF when $t$ is small and $[\boldsymbol{x}]_2$ is large (where $p_t^{(n)}$ has low density).

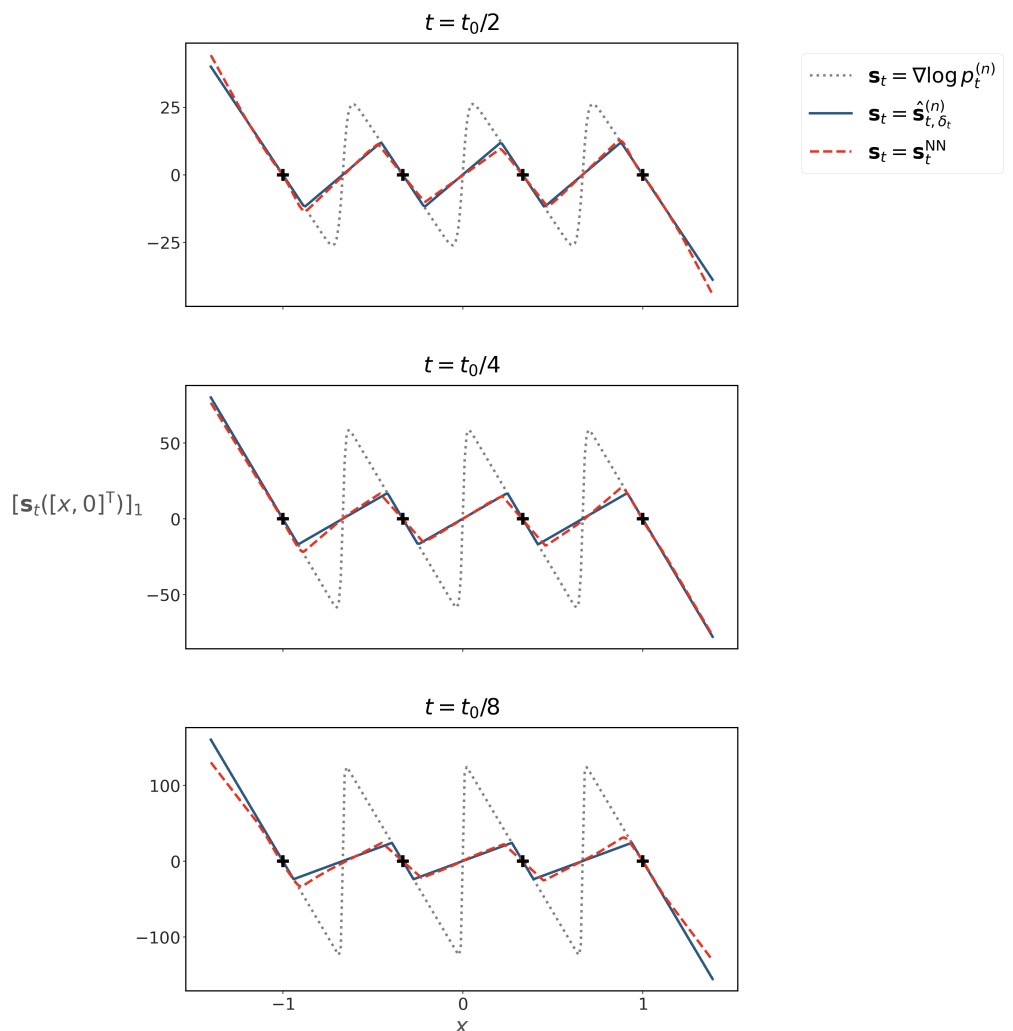

Figure 9: Comparing three SF variants from Experiment 2 in their first (tangent) dimension at different $t$ when they are evaluated on the $[\boldsymbol{x}]_1$ axis. Again, we see a close proximity between the Smoothed PL-ESF and the NN-learned SF, both of which are smoother than the ESF.

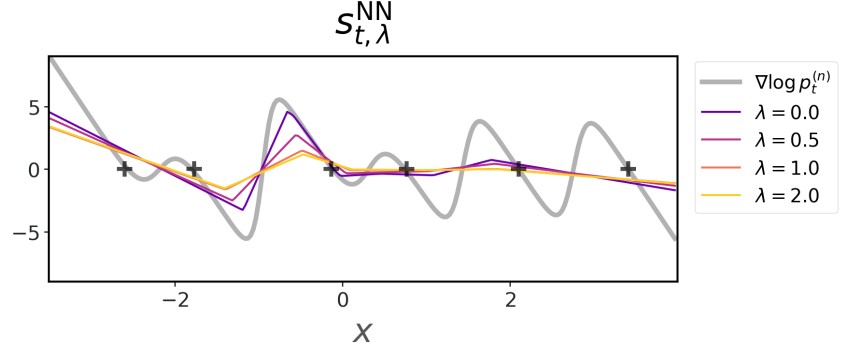

Figure 10: ESF vs NN-learned SF with various strengths of weight decay regularization ($\lambda$) when training data lie non-uniformly in 1-D. We see that the NN-learned SF becomes increasingly smooth as $\lambda$ increases.

## J.4 EXPERIMENTS OF SECTION 6.3

### J.4.1 DATA MANIFOLD AND STEREOGRAPHIC PROJECTIONS

We consider manifolds that are intrinsically 2-D and obtained by slicing the unit sphere in $\mathbb{R}^3$. They can then be embedded in $d > 3$ dimensions by randomly choosing 3-D subspaces within $\mathbb{R}^d$.

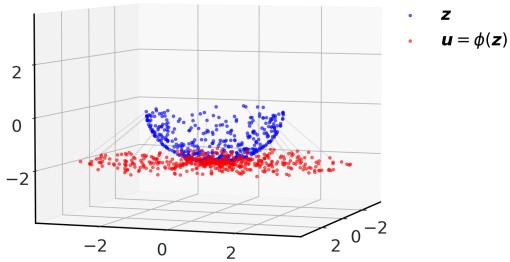

Figure 11: Illustration of the stereographic projection.

The stereographic projection is typically defined as a map in $\mathbb{R}^3$ from the unit sphere (excluding the point on the top) to the x-y plane. By considering concentric spheres with different radii, this can be extended to a bijective map from $\mathbb{R}^3 \setminus \{\boldsymbol{x} \in \mathbb{R}^3 : x_1 = x_2 = 0, x_3 \geq 0\}$ to the lower half of $\mathbb{R}^3$, namely:

$$\Phi(\boldsymbol{x}) = \left( \frac{2\|\boldsymbol{x}\|x_1}{\|\boldsymbol{x}\| - x_3}, \quad \frac{2\|\boldsymbol{x}\|x_2}{\|\boldsymbol{x}\| - x_3}, \quad -\|\boldsymbol{x}\| \right) . \tag{91}$$

This map is illustrated in Figure 11.

### J.4.2 EXPERIMENT SETUP

The two-layer NN has the same architecture as described in Appendix J.2, while the three-layer NN adds one additional layer to the MLP. We consider four scenarios:

1. $d = 3$, $n = 36$, grid in latent space, two-layer NN (Figure 12)
2. $d = 20$, $n = 36$, grid in latent space, two-layer NN (Figure 13)
3. $d = 20$, $n = 36$, grid in latent space, three-layer NN (Figure 14
4. $d = 20$, $n = 64$, uniformly random in latent space, two-layer NN (Figure 6)

where "grid in latent space" means the training set form a uniform grid ($6 \times 6$) in the latent space after the stereographic projection. The models are trained by Adam without weight decay and with learning rate 0.001 with various numbers of epochs.

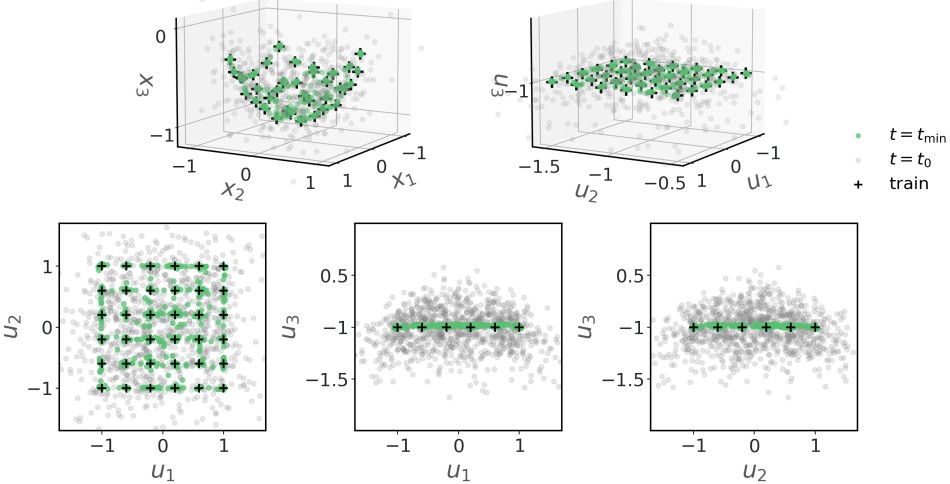

Figure 12: Results of the experiment in Section 6.3, where $d = 3$ and the training set forms a uniform grid in the latent space and the score is learned by a two-layer NN. *Top left*: In the canonical 3-D subspace containing the manifold with coordinates $z = [z_1, z_2, z_3]^{\mathsf{T}}$. *Top right*: Latent coordinates produced by stereographic projection, i.e., $u = \Phi(z)$. *Bottom*: 2-D-sliced views in the latent coordinates.

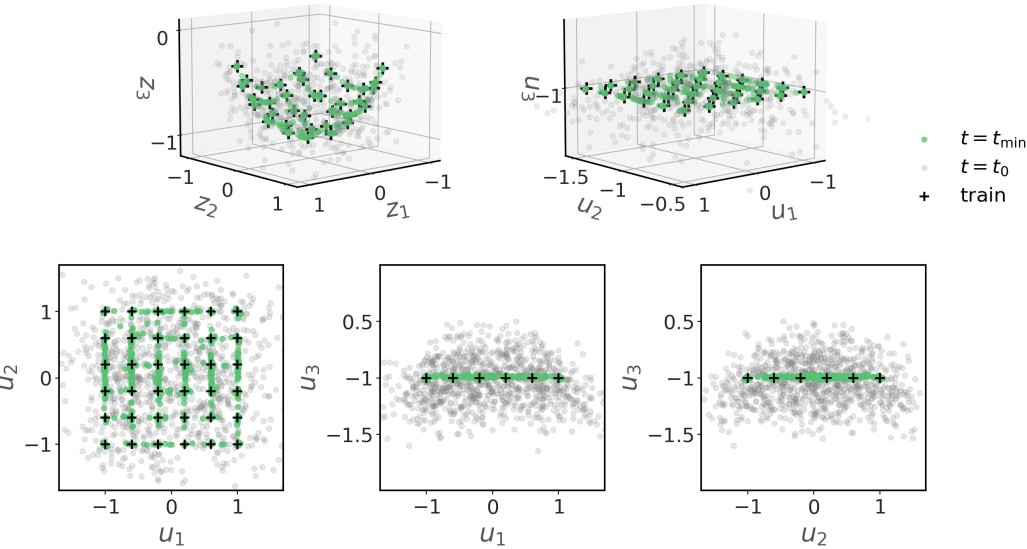

Figure 13: Results of the experiment in Section 6.3, where $d = 20$ and the training set forms a uniform grid in the latent space and the score is learned by a two-layer NN. Same meaning of the plots as in Figure 12.

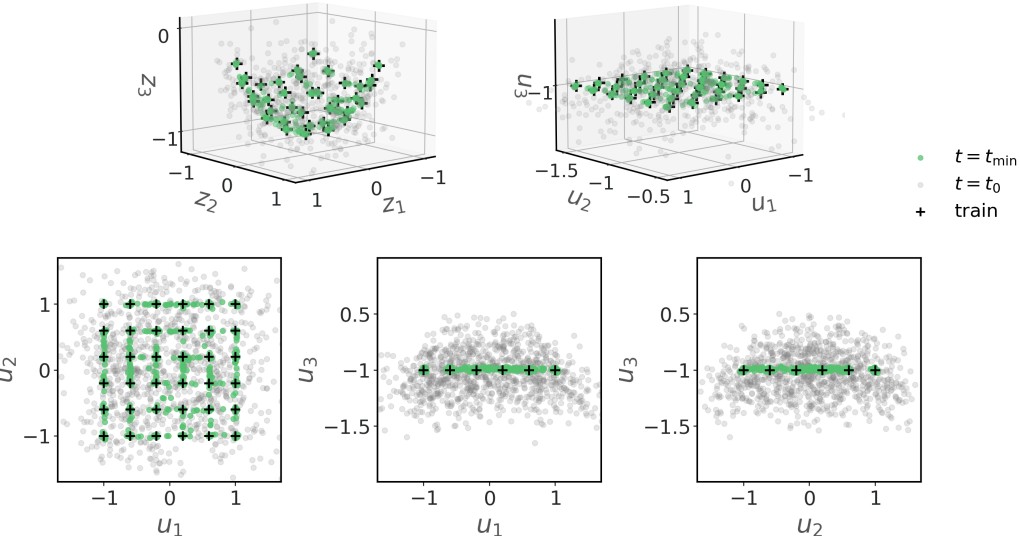

Figure 14: Results of the experiment in Section 6.3, where $d = 20$ and the training set forms a uniform grid in the latent space and the score is learned by a three-layer NN. Same meaning of the plots as in Figure 12.

