# OpenReview forum: "On the Interpolation Effect of Score Smoothing in Diffusion Models"
_ICLR.cc/2026/Conference — ICLR 2026 Poster_

### Official Review · Reviewer_8eUJ · 2025-10-24

**Soundness:** 4
**Presentation:** 2
**Contribution:** 3
**Rating:** 6
**Confidence:** 4

**Summary:**

This paper studies the effect of score smoothing on the ability of a diffusion model to generate samples outside the training data set. Most of the analysis revolves around a simply toy problem with evenly-spaced data points, where the analytic score functions (ESF) are known exactly.  It is shown that the scores learned by a regularized two-layer NN trained on this dataset is closely approximated by a smoothed version of the ESF. In the 1D case, and when the data manifold is a 1D subspace of a larger data space, the smoothed score (which models the NN score) ascribes some probability to regions on the data manifold that lie between the training data samples. At least in this limited setup, smoothed score functions offer a compelling explanation to 'generalization'.

**Strengths:**

1. It is always great to have an analytic model that closely approximates the NN behavior. The authors have done a great job in picking a very minimal setup, modeling the score in closed form, and analyzing it thoroughly to draw their conclusions.

2. The problem being studied is pertinent, and the paper makes clear progress albeit with a simple example.

3. The claims made in the paper are backed up with rigorous proofs.

**Weaknesses:**

1. The core arguments in the paper, which are contained in Sec. 3 and 4, were a little difficult to get through for me. Part of the the problem was that a great deal of notation was introduced in the course of developing the argument that I missed the forest for the trees. I think it would greatly help the paper to have a sketch of the content of these sections precede the more detailed discussion. On a more minor note, making Fig. 2 wider, and stating explicitly that $\lambda$ is the weight decay coefficient around line 161 would also help.

2. The authors use the fact that regularizing the weight norm penalizes non-smooth scores. Later, in Sec. 6.2 they show an experiment where the NN scores smoothly interpolate a circular dataset, even _without weight decay_. This is attributed vaguely to the optimization dynamics. But does this not undercut your main thesis? That is, can you still claim that your smoothed PL ESF reflects the NN scores in more general scenarios where the latter is smoothed by mechanisms other than weight decay? In particular, is there reason to believe that such smoothing will produce a score with smoothness close to that of $r^{*}$? To be clear, this is a question about the _relevance_ of the smoothed PL ESF, rather than the soundness. If there is an answer in the paper that I missed please let me know. Otherwise, it appears to me that this is the primary weakness of the paper.

3. In the extension to higher-dimensions, it would have been interesting to see a data subspace with some curvature, even a mild one. Would some sort of 'locally-flat' assumption help generalize your arguments to such a subspace?

**Questions:**

In proposition 1, if I understood correctly, the function $F^{-1}(\epsilon)$ grows as $\epsilon \to 0$. That means $\kappa$ becomes larger at small $\epsilon$. So $\delta_t = \kappa \sqrt{t}$ is small only when $t$ is very small. But we also know that the actual diffusion model is trained by evaluating the denoising score-matching objective from $t_{\rm min}$ to $T$. If the smoothed PL ESF in Eq. (9) is a 'proxy' for the NN score in your analysis, should you not restrict yourself to using this score only till $t_{\rm min}$. Then, is $\delta_{t_{\rm min}} = \kappa \sqrt{t_{\rm min}}$ necessarily small?

I have given a tentative score based on my current understanding of the paper. I'm willing to update the score if my questions are addressed satisfactorily.

---

> ### Author Response · Authors · 2025-11-20
> **Response to the reviewer**
>
> We thank the reviewer for the helpful and constructive feedback! Please allow us to respond to the comments in detail below.
>
> > W1. The core arguments in the paper, which are contained in Sec. 3 and 4, were a little difficult to get through for me. Part of the the problem was that a great deal of notation was introduced in the course of developing the argument that I missed the forest for the trees. I think it would greatly help the paper to have a sketch of the content of these sections precede the more detailed discussion. On a more minor note, making Fig. 2 wider, and stating explicitly that $\lambda$ is the weight decay coefficient around line 161 would also help.
>
> Thank very much for the helpful suggestions for improving paper presentation. We have incorporated all of them in the updated paper, including adding a paragraph at the end of Section 2 that outlines our main theoretical argument.
>
> > W2. The authors use the fact that regularizing the weight norm penalizes non-smooth scores. Later, in Sec. 6.2 they show an experiment where the NN scores smoothly interpolate a circular dataset, even without weight decay. This is attributed vaguely to the optimization dynamics. But does this not undercut your main thesis? That is, can you still claim that your smoothed PL ESF reflects the NN scores in more general scenarios where the latter is smoothed by mechanisms other than weight decay? In particular, is there reason to believe that such smoothing will produce a score with smoothness close to that of $r^*$? To be clear, this is a question about the relevance of the smoothed PL ESF, rather than the soundness. If there is an answer in the paper that I missed please let me know. Otherwise, it appears to me that this is the primary weakness of the paper.
>
> This is a perceptive observation, and it does **not** undercut the main thesis because even without explicit weight decay, prior works have established theoretically that gradient-based training can induce an **implicit regularization** effect on the learned neural networks in a similar way (the discussion is unfortunately hidden in the Related Works section in Appendix A). Notably, the complexity measure for infinite-width 2-layer homogeneous NNs trained with logistic-type losses exactly agrees with our non-smoothness measure in equation 10 ([1]). Hence, this result allows us to hypothesize that score smoothing may occur through NN training even *without* explicit regularization. We have added a more prominent note on this in Section 3.1.
>
> > W3. In the extension to higher-dimensions, it would have been interesting to see a data subspace with some curvature, even a mild one. Would some sort of 'locally-flat' assumption help generalize your arguments to such a subspace?
>
> We appreciate this insightful suggestion. Indeed, as differentiable manifolds are **locally Euclidean**, they can be approximated locally by linear subspaces with an error dependent on the **curvature**. Hence, our analyses on the interpolation effect on linear subspaces can be applied to an approximate degree under conditions on the curvature. The rigorous theoretical extension of this analysis to curved manifold scenarios is left for future work, as noted in Section 7.
>
> On the empirical side, we have included additional experimental results in the updated paper (Section 6.3) where the training set belongs to 2-dimensional spherical-like manifolds embedded in up to 20 dimensions. This setting allows us to clearly visualize the interpolation phenomenon in higher dimensions via stereographic projections. The results show that, in NN-based diffusion models, the interpolation effect **continues to occur in higher-dimensional settings with nonlinear manifolds** as well as with deeper NN architectures.
>
> > Q1. On the role of $t_{\text{min}}$.
>
> This is an insightful question. Indeed, the analysis works in the regime where $t$ is small relative to the sparsity of the training data ($\Delta$) divided by $\kappa$. Note that in the theoretical analysis, $t_\text{min}$ does **not** appear and we characterize the denoising flow all the way down to time $0$ (cf Propositions 3 and 5 and equations 17 and 18), which is when the distribution collapses exactly onto the subspace. We introduce $t_\text{min}$ only in the experiments to prevent numerical instability due to the increasing magnitude of the score function at small $t$, and it is typically chosen very close to zero. In our experiment in Section 6.1, for example, we choose $t_\text{min} = 10^{-5}$, which is smaller by four orders of magnitude compared to $\Delta / \kappa = (1/3) / 1.2 \approx 0.28$ (Appendix J.2).
>
> Please let us know of any further thoughts and questions. Thanks!
>
> > References
> >
> > 1. Chizat and Bach, “Implicit bias of gradient descent for wide two-layer neural networks trained with the logistic loss.”, COLT 2020

---

> ### Comment · Reviewer_8eUJ · 2025-11-24
> **Follow-up**
>
> Dear authors,
>
> Thanks for your timely response, and for incorporating my feedback. I do have one more question.
>
> I had not read the details in your appendix super carefully, so I did not notice that you used $\Delta$ to quantify training data sparsity. As you pointed out, $t_{\rm min}$ is used to account for 'the increasing magnitude of the score function at small $t$'. However, the denoising score matching loss is exact when the time integral is from $0 \to T$, and it is optimized iff $s_{\theta} = \nabla \log p(x,t)$. *Ideally* we should not need a $t_{\rm min}$.
>
> My intuition is that data sparsity is what necessitates $t_{\rm min}$. That is, at some very small $t$, and around the neighborhood of a training sample $x_i$, the network is learning the score for a delta function at $x_i$, since the score vector in the neighborhood of this point *knows only this point*. The largeness of $s_\theta$ as $t \to 0$ follows from the singular nature of the score of a delta function. Therefore, I would think that if you are doing a proof which accounts for data sparsity (i.e. with $\Delta$ involved), then $t_{\rm min}$ should also enter your derivation in some form.
>
> Or perhaps I am asking a more basic question: shouldn't you be modeling the score that is learned by extremizing the denoising score matching objective $\int^t \mathbb{E}[|| s_\theta(x_t, t) - \nabla \log p(x_t|x_0) ||^2]$ rather than the original intractable object $\int^t \mathbb{E}[|| s_\theta(x_t, t) - \nabla \log p(x_t) ||^2]$? These do not agree at very small $t$ for the reason explained above.

---

> > ### Author Response · Authors · 2025-11-25
> > **Response to the follow-up question**
> >
> > We thank the reviewer for their insightful follow-up questions.
> >
> > > My intuition is that data sparsity is what necessitates $t_{\rm min}$. That is, at some very small $t$, and around the neighborhood of a training sample $x_i$, the network is learning the score for a delta function at $x_i$, since the score vector in the neighborhood of this point knows only this point. The largeness of $s_\theta$ as $t \to 0$ follows from the singular nature of the score of a delta function. Therefore, I would think that if you are doing a proof which accounts for data sparsity (i.e. with $\Delta$ involved), then $t_{\rm min}$ should also enter your derivation in some form.
> >
> > We clarify two possible motivations for introducing a $t_{\rm min}$.
> >
> > 1. **Numerical stability**: We use a small $t_{\rm min}$ (e.g., $t_{\rm eps} = 10^{-5}$ in our experiments) purely to avoid numerical issues in training and inference caused by the singularity of the empirical score function $\nabla \log {p^{(n)}_t}$ as $t \to 0$.
> > 2. **To avoid memorization**: Stopping the backward-in-time denoising dynamics at some $t_{\rm min} > 0$ to prevent it from entering the final regime of collapsing onto the training data points, a strategy that we call “**inference-time early stopping**”. Heuristically, the collapse regime begins when the order of the noise level is on par with $\Delta \gg t_{\rm eps}$.
> >
> > Our work demonstrates that **(explicit or implicit) NN regularization alone is sufficient** to mitigate memorization, and thus $t_{\rm min}$ is needed **only for the first purpose**.
> >
> > Crucially, as we discussed in the paragraph above Section 6, **"inference-time early stopping" does not fundamentally solve memorization**. If the empirical score function $\nabla \log {p^{(n)}\_t}$ is still learned perfectly (i.e., un-regularized and un-smoothed), terminating the denoising dynamics at $t_{\rm min} > 0$ results in the noised empirical distribution $p^{(n)}\_{t\_{\rm min}}$ (equivalent to directly adding noise to the training data) whose support still spans all $d$ dimensions. In contrast, smoothing the empirical score function while still running the denoising dynamics till $t_{\rm eps} \approx 0$ leads to a distribution **whose support recovers the data subspace** without collapsing onto the training data points, which reflects meaningful generalization.
> >
> > Therefore, our theoretical analysis characterizes the denoising flow all the way down to $t=0$ and $t_{\rm min}$ does not appear in the derivation itself.
> >
> > > Or perhaps I am asking a more basic question: shouldn't you be modeling the score that is learned by extremizing the denoising score matching objective $\int^t \mathbb{E}[|| s_{\theta}(x_t, t) - \nabla \log p(x_t | x_0)||^2]$ rather than the original intractable object $\int^t \mathbb{E}[|| s_{\theta}(x_t, t) - \nabla \log p(x_t)||^2]$? These do not agree at very small
> >  for the reason explained above.
> >
> > Note that $|| s_{\theta}(x_t, t) - \nabla \log p(x_t | x_0)||$ in the first term (the DSM objective) is random in both $x_0$ and $x_t$. When the expectation is taken over the random sampling of both $x_0 \sim p_0$ and $x_t \sim p\_{t|0}(x_t | x_0)$, the DSM objective ${\mathbb{E}}\_{x_0, x_t}[|| s\_{\theta}(x_t, t) - \nabla \log p(x_t | x_0)||^2]$ differs from the second term $\mathbb{E}\_{x_t}[|| s\_{\theta}(x_t, t) - \nabla \log p(x_t)||^2]$ only by a term that is **independent from** $\theta$ (as proven in the Appendix of [1]), and hence they are equivalent from the viewpoint of model training. The reviewer is correct that the DSM formulation is the more tractable option in practice.
> >
> > Please correct us in case we misunderstood the original questions and also let us know of any additional thoughts or questions. Thanks!
> >
> > > Reference:
> > >
> > > 1. Vincent, P. (2011). A connection between score matching and denoising autoencoders. Neural computation, 23(7), 1661-1674.

---

> > > ### Comment · Reviewer_8eUJ · 2025-11-25
> > >
> > > Thanks for your response.
> > >
> > > I am aware of the Vincent paper, of course. When I posed the question I was thinking of the score of the 'actual' distribution, not the score of a sum of delta functions over the training samples, what you call $p^{(n)}_0$. The former has no singular behavior at $t=0$, whereas the latter does. For example, if your data distribution is a simple Gaussian $\nabla \log p(x, 0) \propto -x$ but the score $\nabla \log p^{(n)}_0$ will diverge, as in your Eq. (6).
> > >
> > > Your answer clears up my concerns. I have no further questions. Thank you.

---

### Official Review · Reviewer_GL3U · 2025-10-27

**Soundness:** 3
**Presentation:** 3
**Contribution:** 2
**Rating:** 4
**Confidence:** 3

**Summary:**

This work studies how NN-based score predictor can help alleviate the exact memorization phenomenon when an exact score function is used. Specifically, the authors study the score function learned via 2-layer ReLU NN, and show that the learned score function has a smoothing effect that helps creativity.

**Strengths:**

The paper is well-organized and strongly motivated. Studying NN-learned score function is important in understanding the creativity of diffusion model. The theory is rigorous and the authors carry out well-targeted experiments to verify their theoretic results.

**Weaknesses:**

1. missing highly-related prior work [1]

Specifically, in the current paper, the authors mention "while it is difficult to predict exactly what function an NN will learn due to the nonlinearity and stochasticity of the training dynamics" and seeks to find the NN prediction by converting weight decay as a smoothness penalty, while in work [1], the authors already converted the non-convex two-layer NN to a convex program and solved it to optimality, where they showed the optimal function is piecewise linear.

Indeed, Figure 2 in current paper is well-aligned with Figure 1 in [1], where the authors in [1] already theoretically showed why 2-layer ReLU will produce results as Figure 2 in current paper. The 2d circle experiment in current paper was also done in Section 5.2 in [1]. Also, formula (9) in current paper looks similar to formula (7) in [1].

The highly-correlated results and scientific problems being studied should be paid attention.

2. the setting is limited to 2 layer ReLU, and is far from what's being used in real diffusion model


[1] Analyzing Neural Network-Based Generative Diffusion Models through Convex Optimization: https://arxiv.org/abs/2402.01965

**Questions:**

1. is there any clue how to extend the current analysis framework to deeper NN architecture?

2. if it's hard to generalize current result to modern architecture, is there at least any clue that whether modern diffusion models' effectiveness also comes from such score smoothing mechanism?

3. if the authors take a look at the prior work mentioned above, there is some deeper connections between these two (which might also be superficial), like the current work has sign(x) for 1d data, and the derived convex program also has sign(x) (see the paragraph below Theorem 3.1). Is this coincidence?

---

> ### Author Response · Authors · 2025-11-20
> **Response to the reviewer**
>
> We thank the reviewer for the helpful and constructive feedback! Please allow us to respond to the comments in detail below.
>
> > W1. On the relationship with prior work [1]
>
> We sincerely thank the reviewer for bringing this excellent work to our attention and apologize for its omission in our initial literature search. We have added a comprehensive discussion of [1] in the Related Works section. The work of [1] builds on a valuable insight on the equivalence between (i) optimizing two-layer ReLU-like neural networks (NNs) on **finite data** and (ii) a convex optimization problem in a transformed dual space, and the authors leverage this insight to derive variational forms of the NN-based score function learning.
>
> However, a **key difference** in the problem formulation distinguishes our efforts. The authors of [1] introduce a **non-trivial simplification** in their empirical denoising score matching (DSM) objective (their Equation 8), where the expectation over Gaussian noise is approximated by sampling a single noise vector per training data point. This approximation reduces the problem to a finite-data setting, under which the convex program formulation becomes possible, but concurrently introduces **variance** to  the problem and therefore its solutions. In contrast, our score matching loss (Equation 4) is defined with the expectation taken over the noised empirical distribution ($p_t^{(n)}$), which is **not** decomposable into a sum over finite data points and makes the convex program formulation from [1] **inapplicable**.
>
> (*Note*: Their work also considers an alternative score matching objective - their Equation 1 - defined with the Jacobian trace and averaged over the training set. But in the context of denoising diffusion, the relevant objective should have the expectation taken over the noised empirical distribution (our $p^{(n)}_t$) rather than just the training set, and this would similarly make the convex program formulation inapplicable.)
>
> ***Comparison of score estimators and generated distributions***: Because of the different problem setup, our results are fundamentally different even though they may appear similar. For example, while the 1D score predictor from Equation 7 and Figure 1 in [1] is also piecewise linear, it is **fully linear** within the range of data samples, whereas our Smoothed PL score estimator changes its slope twice between every pair of data points (note that our Figure 2 is visually similar because it considers the case with two training data points; Figure 8 (9 in the updated version) shows the $n=4$ setting). The difference is also reflected in the generated distribution: their score estimator produces **Gaussian or Gauss-Laplace distributions** (their Theorem 3.3), while ours drives the denoising dynamics toward distributions on **desired subspaces that interpolate the training data**.
>
> ***Comparison of experiments***: The experiments in [1] use their convex-program-based score estimator and **do not involve actual score functions learned by NNs**. In contrast, our experiments in Section 6.2 provide crucial empirical evidence showing that **NN-learned score functions** (i) exhibit a smoothing effect and (ii) enable the denoising dynamics to produce samples that interpolate the training set in a manner that resembles the 1D scenario.
>
> ***Re: “in work [1], the authors already converted the non-convex two-layer NN to a convex program and solved it to optimality”***: The thesis from [1] that 'the NN training objective is equivalent to a convex program' only holds **in terms of their global minima**.  Since there is no guarantee that gradient-based NN training will find this global minimum, [1]’s result is similarly limited in power due to the inherent non-convexity and stochasticity of NN training dynamics.

---

> > ### Author Response · Authors · 2025-11-20
> > **Response to the reviewer (continued)**
> >
> > > W2. "The setting is limited to 2 layer ReLU, and is far from what's being used in real diffusion model"
> > >
> > > Q1. "Is there any clue how to extend the current analysis framework to deeper NN architecture?"
> >
> > This is an excellent question. On the theoretical front, we think a generalization is possible if further progress is made towards better characterizing the **complexity measures** associated with deeper NN architectures (e.g. [2, 3]). Given that this is a non-trivial challenge largely orthogonal to our main thesis, we are deferring this effort to future work.
> >
> > Empirically, however, our latest experiment results (added in Section 6.3 and Appendix J.4) demonstrate that the interpolation effect persists in diffusion models with **deeper NN architectures** as the score estimator as well as in **higher-dimensional settings with nonlinear manifolds**.
> >
> > > Q2. "If it's hard to generalize current result to modern architecture, is there at least any clue that whether modern diffusion models' effectiveness also comes from such score smoothing mechanism?"
> >
> > A relevant study is [4], which demonstrates empirically with real-world **image datasets** that **UNet**-based diffusion models can generate samples that interpolate between different modes in the training distribution, and the authors considered this a mechanism behind *hallucination* behaviors. We think the story for *generalization* is similar, as the conceptual difference between generalization and hallucination lies in whether interpolation occurs within or outside of the support of the data distribution. Further elucidation of this connection is an exciting direction for future research.
> >
> > > Q3. "If the authors take a look at the prior work mentioned above, there is some deeper connections between these two (which might also be superficial), like the current work has sign(x) for 1d data, and the derived convex program also has sign(x) (see the paragraph below Theorem 3.1). Is this coincidence?"
> >
> > Yes, they arise from different origins. In [1], the “sign(x)” terms are linked to the **activation patterns** of the ReLU / Abs activation functions. In our work, “sign(x)” appears when we analyze the empirical score function (ESF) in the low noise limit, a phenomenon that exists **before** NN learning is introduced.
> >
> > Please let us know of any further thoughts and questions. Thanks!
> >
> > >References:
> > >
> > >1. Zhang and Pilanci, "Analyzing Neural Network-Based Generative Diffusion Models through Convex Optimization", NeurIPS OPT Workshop 2025
> > >
> > >2. E et al., “The Barron Space and the Flow-Induced Function Spaces for Neural Network Models”, Constructive Approximation (2022)
> > >
> > >3. Chen, “Neural Hilbert Ladders: Multi-Layer Neural Networks in Function Space”, JMLR (2024)
> > >
> > >4. Aithal et al., “Understanding Hallucinations in Diffusion Models through Mode Interpolation”, NeurIPS 2024

---

### Official Review · Reviewer_1ab2 · 2025-10-31

**Soundness:** 3
**Presentation:** 3
**Contribution:** 3
**Rating:** 6
**Confidence:** 3

**Summary:**

This paper studies why score-based diffusion models (DMs) with neural networks can generate new samples instead of simply memorizing their training data.
Besides, the authors propose a mathematically tractable Smoothed Piecewise-Linear Empirical Score Function and show that this smoothing prevents the reverse-time denoising process from collapsing to training points.
Through analytical results and simple experiments, they demonstrate that both smoothed and NN-learned score functions produce smooth interpolations rather than pure memorization.

**Strengths:**

1. The paper gives a clear and rigorous mathematical formulation of how score smoothing prevents collapse and enables interpolation in diffusion models.
2. It provides elegant geometric insight and connects explicit smoothing with the implicit regularization effect of neural network training.

**Weaknesses:**

1. The paper does not theoretically prove that neural network training truly realizes the same smoothing mechanism.
2. All experiments are conducted on low-dimensional toy cases, and it remains uncertain whether the proposed mechanism can actually help reduce memorization in diffusion models trained on limited real-world data.

**Questions:**

See Weakness. Further:
  1. Can the proposed smoothing framework be extended or tested on high-dimensional real data ？

---

> ### Author Response · Authors · 2025-11-20
> **Response to the reviewer**
>
> We thank the reviewer for the helpful and constructive feedback! Please allow us to respond to the questions and comments here.
>
> > W1. The paper does not theoretically prove that neural network training truly realizes the same smoothing mechanism.
>
> The assessment is correct: our result does not go as far as guaranteeing that **finite-width neural networks trained by gradient-based algorithms** will *necessarily* learn the same solution as the Smoothed Piece-wise Linear (PL) score estimator. Rigorously doing so requires tackling the well-known challenge of **non-convexity** in neural network (NN) optimization, which is a complex topic beyond the scope of the current paper.
>
> However, we *did* establish a **significant connection** between the Smoothed PL score and NN learning: the former is nearly optimal in terms of minimizing the penalty term in equation 10 – known to be equivalent to **explicit and implicit regularization in two-layer ReLU neural networks** - while attaining a low score matching loss (discussed in Section 3.1). To our knowledge, our work is the first to attempt at deriving approximate solutions to such type of functional-space variational problems in concrete settings, which furthermore yields direct implications for phenomena in learning - namely, the memorization behavior of diffusion models.
>
>
> > W2. All experiments are conducted on low-dimensional toy cases, and it remains uncertain whether the proposed mechanism can actually help reduce memorization in diffusion models trained on limited real-world data.
> >
> > Q. Can the proposed smoothing framework be extended or tested on high-dimensional real data?
>
> In the updated paper (Section 6.3), we have included additional experimental results specifically targeting higher-dimensional data, where the training set belongs to 2-dimensional spherical-like manifolds embedded in up to 20 dimensions. This setting allows us to clearly visualize the interpolation phenomenon in higher dimensions via stereographic projections. The results show that, in NN-based diffusion models, the interpolation effect **persists in higher-dimensional settings with nonlinear manifolds** as well as with deeper NN architectures.
>
> On the theoretical front, we believe the framework can be extended to high-dimensional manifold settings with assumptions on the **curvature** of the underlying manifold. Because differentiable manifolds are locally Euclidean, they can be approximated locally by linear subspaces with an error dependent on the curvature, and hence our analyses on the interpolation effect based on linear subspaces can be applied to an approximate degree. We leave the rigorous extension of the theoretical analysis to curved manifold scenarios to future work.
>
> Please let us know of any further thoughts and questions. Thanks!

---

### Official Review · Reviewer_PEv5 · 2025-11-04

**Soundness:** 3
**Presentation:** 3
**Contribution:** 3
**Rating:** 6
**Confidence:** 4

**Summary:**

This paper studies the interpolation effect of NN-learned score estimator in diffusion models. Numerical experiments are conducted to support theoretical analysis.

**Strengths:**

1. By using a two-point 1-d example, the authors analyze properties of a smoothed piece-wise linear score estimator.
2. The authors investigates the evolution of marginal distribution in sampling using the above estimator
3. For multi-point case, the theory is also established.
4. Numerical experiments are conducted for this special estimator with comparison to neural networks.

**Weaknesses:**

I have several major concerns:
1. Essentially, the authors constructed a special score estimator and argues diffusion model sampling benefits from this estimator. However, it is not clear to me how this estimator is connected to neural networks. Even in 1-d two-point case, such a connection is missing.
2. Seemingly the authors propose a new regularized score estimator to make it smoother and thus benefiting the sampling process. However, in practice, the regularized training objective (10) and (20) seem to be hard to compute. Could you elaborate this point further?
3. In terms of experiments, the NN-learned estimator indeed behave similarly to the one proposed by authors. Is it because that NN learns exactly this piece-wise linear form or NN estimator enjoys some other smoothing properties?

**Questions:**

I have a minor question:
1. Although maybe not easy, I would appreciate it if authors could show numerical results for high-dimensional data.

---

> ### Author Response · Authors · 2025-11-20
> **Response to the reviewer**
>
> We thank the reviewer for the helpful and constructive feedback! Please allow us to respond to the questions and comments here.
>
> *General note*: We would like to clarify the role of the Smoothed Piece-wise Linear (PL) score estimator in our work: it is **not** proposed as a practical score estimator, but rather a **theoretical construct** designed for analyzing the smoothing mechanism in NN-learned score estimators, thanks to its **connection to the complexity measure associated with two-layer NNs**. We address the individual comments in detail below.
>
>
> > W1. Essentially, the authors constructed a special score estimator and argues diffusion model sampling benefits from this estimator. However, it is not clear to me how this estimator is connected to neural networks. Even in 1-d two-point case, such a connection is missing.
>
> The connection between the Smoothed PL estimator and NNs is elaborated in the paragraph preceding equation 10. Specifically, prior work ([1]) has established that training a two-layer ReLU NN with unlimited width and regularized weight norms is essentially **equivalent to function fitting with the non-smoothness penalty** defined by equation 10. Furthermore, later results ([2]) demonstrate that the same effective regularizer can arise from **implicit regularization** during the gradient descent training of two-layer NNs, even without explicit weight decay. We have added a further note on this in Section 3.1.
>
> > W2. Seemingly the authors propose a new regularized score estimator to make it smoother and thus benefiting the sampling process. However, in practice, the regularized training objective (10) and (20) seem to be hard to compute. Could you elaborate this point further?
>
> As stated, the regularizers here are **not** intended for practical application. Their sole purpose is to serve as a crucial intermediary in the theoretical understanding of how smoothing occurs in NN-learned score estimators.
>
> > W3. In terms of experiments, the NN-learned estimator indeed behave similarly to the one proposed by authors. Is it because that NN learns exactly this piece-wise linear form or NN estimator enjoys some other smoothing properties?
>
> Indeed, these results show that the Smoothed PL score serves as a **good proxy** for what the NN learns, thus providing compelling **empirical validation** for the theoretical link between NN learning and the Smoothed PL score estimator discussed above.
>
> > Q1. Although maybe not easy, I would appreciate it if authors could show numerical results for high-dimensional data.
>
> We have incorporated additional experimental results in the updated paper (Section 6.3) that specifically address high-dimensional data. The training data belong to 2-dimensional curved manifolds embedded in up to 20 dimensions, chosen so that the interpolation phenomenon can be well-visualized through stereographic projections. The results demonstrate that **the interpolation effect persists in higher-dimensional settings and with non-linear data manifolds**.
>
> Please let us know of any further thoughts and questions. Thanks!
>
> > References
> >
> > 1. Savarese et al., “How do infinite width bounded norm networks look in function space?”, COLT 2019.
> >
> > 2. Chizat and Bach, “Implicit bias of gradient descent for wide two-layer neural networks trained with the logistic loss.”, COLT 2020

---

### Meta-Review · Area_Chair_gChQ · 2025-12-31

**Summary:**

This paper offers a clear and timely hypothesis for why score-based diffusion models can generate novel samples: apparent creativity emerges from an interpolation effect induced by smoothing the empirical score. By focusing on an analytically tractable setting where training data lie uniformly on a 1D subspace, the authors disentangle how score smoothing interacts with denoising dynamics, supported by both closed-form analysis and numerical validation. The main contribution is a convincing demonstration that smoothed scores can generate samples that interpolate within the data subspace—capturing structure without collapsing into exact memorization. A second, practically relevant takeaway is the evidence that standard neural-network regularization can mimic this smoothing effect, extending (at least in simple regimes) beyond purely linear settings.

Overall, the work is well-motivated and technically sound in its core setting, and it connects a mechanistic explanation (score smoothing) to observable generation behavior (interpolation vs. memorization). It makes meaningful contributions to the machine learning community in a timely manner.

**Reviewer Concerns:**

1. One major concern is the connection with real-world neural networks. The authors acknowledged the theoretical limitations and addressed them through experimental results, corroborating their claims.

1. Another main limitation is the narrowness of the theoretical regime (uniform 1D subspace), which raises questions about how directly the conclusions transfer to realistic high-dimensional data distributions. The reviewer partially addressed this through experiments on high-dimensional data, corroborating his claims.

2. A minor concern is about the incomplete literature review. The authors have addressed this by acknowledging and discussing the literature more comprehensively.

**Reviewer Scores:**

Reviewer GL3U would raise his score:
The authors comprehensively addressed the concerns from Reviewer GL3U.

---

### Decision · Program_Chairs · 2026-01-26

Accept (Poster)